# On the Tension between Byzantine Robustness and No-Attack Accuracy in Distributed Learning

Yi-Rui Yang [1]   Chang-Wei Shi [1]   Wu-Jun Li [1]

## Abstract

Byzantine-robust distributed learning (BRDL), which refers to distributed learning that can work with potential faulty or malicious workers (also known as Byzantine workers), has recently attracted much research attention. Robust aggregators are widely used in existing BRDL methods to obtain robustness against Byzantine workers. However, Byzantine workers do not always exist in applications. As far as we know, there is almost no existing work theoretically investigating the effect of using robust aggregators when there are no Byzantine workers. To bridge this knowledge gap, we theoretically analyze the aggregation error for robust aggregators when there are no Byzantine workers. Specifically, we show that the worst-case aggregation error without Byzantine workers increases with the increase of the number of Byzantine workers that a robust aggregator can tolerate. The theoretical result reveals the tension between Byzantine robustness and no-attack accuracy, which refers to accuracy without faulty workers and malicious workers in this paper. Furthermore, we provide lower bounds for the convergence rate of gradient descent with robust aggregators for non-convex objective functions and objective functions that satisfy the Polyak-Łojasiewicz (PL) condition, respectively. We also prove the tightness of the lower bounds. The lower bounds for convergence rate reveal similar tension between Byzantine robustness and no-attack accuracy. Empirical results further support our theoretical findings.

[1]National Key Laboratory for Novel Software Technology, School of Computer Science, Nanjing University, Nanjing, China. Correspondence to: Wu-Jun Li <liwujun@nju.edu.cn>.

*Proceedings of the 42ⁿᵈ International Conference on Machine Learning*, Vancouver, Canada. PMLR 267, 2025. Copyright 2025 by the author(s).

## 1. Introduction

Distributed learning has been a hot research topic for years (Haddadpour et al., 2019; Jaggi et al., 2014; Lee et al., 2017; Lian et al., 2017; Ma et al., 2015; Shamir et al., 2014; Sun et al., 2018; Yang, 2013; Yu et al., 2019a;b; Zhao et al., 2017; 2018; Zhou et al., 2018; Zinkevich et al., 2010). Traditional distributed learning typically assumes no attack or failure. However, in some real-world scenarios such as federated learning (McMahan & Ramage, 2017), the probability of attacks or failure greatly increases due to the server's weak control on workers (Baruch et al., 2019; Kairouz et al., 2021; Xie et al., 2020a). The workers under attack or failure are also called Byzantine workers (Lamport et al., 1982). Byzantine-robust distributed learning (BRDL), which refers to distributed learning that can work with potential Byzantine workers, has recently attracted much research attention (Bernstein et al., 2019; Bulusu et al., 2021; Chen et al., 2018; Damaskinos et al., 2018; Diakonikolas et al., 2017; Diakonikolas & Kane, 2019; Konstantinidis & Ramamoorthy, 2021; Rajput et al., 2019; Sohn et al., 2020; Wu et al., 2020; Yang & Li, 2021; Yang et al., 2024; 2020; Yin et al., 2019).

Robust aggregation is a widely used technique to obtain Byzantine robustness in distributed learning. By replacing the vanilla mean aggregator on the server with some robust aggregators, the distributed learning method can be robust against a certain number of Byzantine workers. There are many robust aggregators proposed in existing works (Blanchard et al., 2017; Chen et al., 2017; Guerraoui et al., 2018; Karimireddy et al., 2021; Yin et al., 2018). Meanwhile, using robust aggregation will also inevitably introduce an aggregation error, which is the square distance between the aggregation result and the true mean value. Large aggregation error will lead to a decrease of model accuracy.

Existing analyses (Allouah et al., 2023; Karimireddy et al., 2021; 2022) for the aggregation error of robust aggregators typically assume that the number (or fraction) of Byzantine workers is known in advance, while in real-world applications, the number of Byzantine workers is usually unavailable. It is advised in existing works (Karimireddy et al., 2022; Yang & Li, 2023) to determine the maximum number of Byzantine workers that the BRDL method can tolerate

in advance and then set the robust aggregator accordingly. However, Byzantine workers do not always exist in applications. When there are actually no Byzantine workers, the aggregation error introduced by robust aggregators could have a negative effect on the convergence of distributed learning methods and degrade the model accuracy. As far as we know, there is almost no existing work theoretically investigating the performance of robust aggregators when there are no Byzantine workers. To bridge this knowledge gap, we mainly focus on the following question:

*How large could the aggregation error be when there are actually no Byzantine workers?*

The main contributions of our work are listed as follows:

- To the best of our knowledge, this is the first work that theoretically investigates the tension between Byzantine robustness and no-attack accuracy in distributed learning.

- We theoretically prove that the worst-case aggregation error without Byzantine workers increases with the increase of $f$, which is the number of Byzantine workers that a robust aggregator can tolerate. The theoretical result reveals the tension between Byzantine robustness and no-attack accuracy.

- Moreover, we provide lower bounds for the convergence rate of gradient descent with robust aggregators for non-convex objective functions and objective functions that satisfy the Polyak-Łojasiewicz (PL) condition, respectively. We further show that the lower bounds are tight and increase with the increase of $f$.

- We also provide empirical results, which further support our theoretical findings.

In addition, we would like to point out that existing works on robust aggregators mainly explore the performance when there exist Byzantine workers while we focus on the case without Byzantine workers in this work.

## 2. Preliminary

Many distributed learning problems can be formulated as:

$$\min_{\mathbf{w} \in \mathbb{R}^d} F(\mathbf{w}) = \frac{1}{n} \sum_{i=1}^{n} F_i(\mathbf{w}), \tag{1}$$

where $\mathbf{w}$ is the model parameter, $n$ is the number of workers and $F_i(\mathbf{w})$ is the loss function associated with the training data on the $i$-th worker. Moreover, we mainly focus on the Parameter Server (PS) framework, where there is an extra server responsible for coordination. A widely used method

---

**Algorithm 1** Byzantine-Robust Gradient Descent (ByzGD)

**Input:** iteration number $T$, learning rates $\{\eta_t\}_{t=0}^{T-1}$, robust aggregator $\mathbf{Agg}(\cdot)$;
**Initialization:** model parameter $\mathbf{w}_0$;
**for** $t = 0$ **to** $T - 1$ **do**
 Broadcast $\mathbf{w}_t$ to all workers;
 **on worker** $i \in \{1, \ldots, n\}$ **in parallel do**
  Compute local gradient $\mathbf{g}_i = \nabla F_i(\mathbf{w}_t)$;
  Send $\mathbf{g}_i$ to the server;
 **end on worker**
 Compute: $\mathbf{w}_{t+1} = \mathbf{w}_t - \eta_t \cdot \mathbf{Agg}(\mathbf{g}_1, \ldots, \mathbf{g}_n)$;
**end for**
**Output:** model parameter $\mathbf{w}_T$.

---

to solve the problem (1) is gradient descent as presented below:

$$\mathbf{w}_{t+1} = \mathbf{w}_t - \eta_t \cdot \left[ \frac{1}{n} \sum_{i=1}^{n} \mathbf{g}_i \right].$$

Specifically, at the $t$-th iteration, all the workers compute local gradients $\mathbf{g}_1, \ldots, \mathbf{g}_n$ in parallel and send them to the server. Then, the server aggregates the gradients with vanilla averaging and updates the model parameter with the aggregated gradient. However, the averaging value is sensitive to the outliers and not robust against Byzantine workers. Therefore, robust aggregators such as Krum (Blanchard et al., 2017), geometric median (Chen et al., 2017) and coordinate-wise trimmed mean (Yin et al., 2018) are proposed to replace vanilla averaging in BRDL. By replacing the vanilla averaging with some robust aggregators, we can obtain Byzantine-robust gradient descent (ByzGD). In ByzGD, the model parameters are updated by:

$$\mathbf{w}_{t+1} = \mathbf{w}_t - \eta_t \cdot \mathbf{Agg}(\mathbf{g}_1, \ldots, \mathbf{g}_n),$$

where $\mathbf{Agg}(\cdot)$ denotes a general robust aggregator. More details about ByzGD are presented in Algorithm 1.

In existing works, there are several definitions of robust aggregators that are used to analyze the aggregation error. The definition of $(\delta_{\max}, c)$-*ARAgg* (Karimireddy et al., 2022) is proposed based on the expectation of the distances among the vectors while the definition of $(f, \lambda)$-*resilient averaging* (Farhadkhani et al., 2022) is proposed based on the maximum distances among the vectors. In (Allouah et al., 2023), the definition of $(f, \kappa)$-*robust aggregator* is further proposed, which can unify many existing definitions of robust aggregators including $(\delta_{\max}, c)$-ARAgg and $(f, \lambda)$-resilient averaging. Due to this reason, we mainly follow the definition of $(f, \kappa)$-robust aggregator in this work. For simplicity, we use the notation $\|\cdot\|$ to represent the $L_2$-norm in this paper.

**Definition 2.1** $((f, \kappa)$-robustness (Allouah et al., 2023))**.** Let $f < \frac{n}{2}$ and $\kappa \geq 0$. An aggregator $\mathbf{Agg} : \mathbb{R}^{d \times n} \to \mathbb{R}^d$ is

said to be $(f, \kappa)$-robust if for any $n$ vectors $\mathbf{x}_1, \ldots, \mathbf{x}_n \in \mathbb{R}^d$ and for any set $S \subseteq \{1, \ldots, n\}$ satisfying $|S| = n - f$, we have

$$\|\mathbf{Agg}(\mathbf{x}_1, \ldots, \mathbf{x}_n) - \bar{\mathbf{x}}_S\|^2 \leq \kappa \cdot \left[ \frac{1}{|S|} \sum_{i \in S} \|\mathbf{x}_i - \bar{\mathbf{x}}_S\|^2 \right],$$

where $\bar{\mathbf{x}}_S = \frac{1}{|S|} \sum_{i \in S} \mathbf{x}_i$.

For an $(f, \kappa)$-robust aggregator, the square distance between the aggregated result $\mathbf{Agg}(\mathbf{x}_1, \ldots, \mathbf{x}_n)$ and the mean value of non-Byzantine workers $\bar{\mathbf{x}}_S$ is bounded. $\|\mathbf{Agg}(\mathbf{x}_1, \ldots, \mathbf{x}_n) - \bar{\mathbf{x}}_S\|^2$ is also known as the aggregation error. Since the server cannot identify Byzantine workers, the inequality should be satisfied for any set $S$ with $|S| = n - f$, where $f$ is the number of Byzantine workers that the aggregator can tolerate. Specifically, when there are actually no Byzantine workers, the aggregation error is $\|\mathbf{Agg}(\mathbf{x}_1, \ldots, \mathbf{x}_n) - \frac{1}{n} \sum_{i=1}^{n} \mathbf{x}_i\|^2$. Thus, we propose the definition of $\epsilon$-accuracy in Definition 2.2, which measures the aggregation error when there are no Byzantine workers.

**Definition 2.2** ($\epsilon$-accuracy). Let $\epsilon \geq 0$. An aggregator $\mathbf{Agg} : \mathbb{R}^{d \times n} \to \mathbb{R}^d$ is said to be $\epsilon$-accurate if for any $n$ vectors $\mathbf{x}_1, \ldots, \mathbf{x}_n \in \mathbb{R}^d$, we have

$$\|\mathbf{Agg}(\mathbf{x}_1, \ldots, \mathbf{x}_n) - \bar{\mathbf{x}}\|^2 \leq \epsilon \cdot \left[ \frac{1}{n} \sum_{i=1}^{n} \|\mathbf{x}_i - \bar{\mathbf{x}}\|^2 \right],$$

where $\bar{\mathbf{x}} = \frac{1}{n} \sum_{i=1}^{n} \mathbf{x}_i$.

Comparing Definition 2.1 and 2.2, we can find that $\epsilon$-accuracy is equivalent to $(f, \kappa)$-robustness when $f = 0$ and $\kappa = \epsilon$. However, it is meaningless to say the robustness of an aggregator that can tolerate at most 0 Byzantine worker ($f = 0$). Thus, for better readability, we present the definition of $\epsilon$-accuracy alone, which is about the performance of an aggregator in the case without Byzantine workers.

## 3. Analysis of Aggregation Error

In this section, we theoretically analyze the aggregation error of $(f, \kappa)$-robust aggregators in the case without Byzantine workers. We only present the main results and proof sketches in this section. Proof details are deferred to Appendix A in the supplementary material due to the limited space.

To avoid confusion, we would like to clarify that $f$ is the number of Byzantine workers that the aggregator can tolerate and there are actually no Byzantine workers in the case that we consider. Specifically, we will focus on how large $\epsilon$ (please refer to Definition 2.2) could be in the case without Byzantine workers.

### 3.1. General Lower Bound

Firstly, we provide a lower bound of $\epsilon$ for $(f, \kappa)$-robust aggregators.

**Theorem 3.1** (Lower Bound). *If an $(f, \kappa)$-robust aggregator is $\epsilon$-accurate, we have $\epsilon \geq \frac{f}{n-f}$.*

The lower bound $\frac{f}{n-f}$ in Theorem 3.1 is between $0$ and $1$ since $f < \frac{n}{2}$. Moreover, when the total number of workers $n$ is fixed, the lower bound increases as $f$ increases. In other words, making the aggregator robust to more Byzantine workers will inevitably introduce a larger worst-case aggregation error. We will discuss more about this after proving the tightness of the lower bound in Theorem 3.1.

### 3.2. General Upper Bound

Then we provide a general upper bound of $\epsilon$ for all $(f, \kappa)$-robust aggregators in Theorem 3.2.

**Theorem 3.2.** *Any $(f, \kappa)$-robust aggregator is $\kappa$-accurate.*

The constant $\kappa$ is dependent on $f$ and differs for different robust aggregators (Allouah et al., 2023). We present the values of $\kappa$ for three common $(f, \kappa)$-robust aggregators, which are known as coordinate-wise trimmed mean (TM), coordinate-wise median (CM) and geometric median (GM), respectively, and the lower bound of $\kappa$ in Table 1.

As we can see, the values of $\kappa$ are much larger than the lower bound $\frac{f}{n-f}$ of $\epsilon$. Specifically, for GM and CM, the value of $\kappa$ is larger than $4$ while the lower bound $\frac{f}{n-f}$ of $\epsilon$ is always smaller than $1$ when $f < \frac{n}{2}$. Moreover, even the lower bound $\frac{f}{n-2f}$ of $\kappa$ will be much larger than $\frac{f}{n-f}$ when $f$ is close to $\frac{n}{2}$ since the denominator is close to $0$. To obtain tighter results, then we separately analyze the aggregation error of each aggregator.

### 3.3. Analysis for Specific Aggregators

In this section, we analyze the value of $\epsilon$ for TM, CM, and GM. Firstly, we present the definitions of these three common $(f, \kappa)$-robust aggregators.

**Definition 3.3** (coordinate-wise trimmed mean (Yin et al., 2018)). Let vectors $\mathbf{x}_1, \ldots, \mathbf{x}_n \in \mathbb{R}^d$. The coordinate-wise trimmed mean $TM_{f/n} : \mathbb{R}^{d \times n} \to \mathbb{R}^d$ is defined as:

$$[TM_{f/n}(\mathbf{x}_1, \ldots, \mathbf{x}_n)]_j = \frac{1}{n - 2f} \sum_{x \in \mathcal{X}_j} x,$$

where $[\cdot]_j$ denotes the $j$-th coordinate of a vector and the set $\mathcal{X}_j$ is obtained by removing the $f$ largest and the $f$ smallest values of $\{[\mathbf{x}_1]_j, \ldots, [\mathbf{x}_n]_j\}$.

**Definition 3.4** (coordinate-wise median (Yin et al., 2018)). Let $\mathbf{x}_1, \ldots, \mathbf{x}_n \in \mathbb{R}^d$. The coordinate-wise median $CM$ :

*Table 1.* Values of $\kappa$ for different robust aggregators (Allouah et al., 2023)

| Aggregator | TM | CM | GM | Lower bound |
|---|---|---|---|---|
| $\kappa$ | $\frac{6f}{n-2f}(1+\frac{f}{n-2f})$ | $4(1+\frac{f}{n-2f})^2$ | $4(1+\frac{f}{n-2f})^2$ | $\frac{f}{n-2f}$ |

*Table 2.* Values of $\epsilon$ for different robust aggregators

| Aggregator | $\text{TM}_{f/n}$ | CM | GM | Lower bound |
|---|---|---|---|---|
| $\epsilon$ | $\frac{f}{n-f}$ | $\frac{\lfloor\frac{n-1}{2}\rfloor}{n-\lfloor\frac{n-1}{2}\rfloor}$ | $1$ | $\frac{f}{n-f}$ |

$\mathbb{R}^{d\times n} \to \mathbb{R}^d$ is defined as:

$$[CM(\mathbf{x}_1,\ldots,\mathbf{x}_n)]_j = Median([\mathbf{x}_1]_j,\ldots,[\mathbf{x}_n]_j),$$

where $Median(\cdot)$ is the scalar median operator.

**Definition 3.5** (geometric median). Let $\mathbf{x}_1,\ldots,\mathbf{x}_n \in \mathbb{R}^d$. The geometric median $GM : \mathbb{R}^{d\times n} \to \mathbb{R}^d$ is defined as:

$$GM(\mathbf{x}_1,\ldots,\mathbf{x}_n) \in \arg\min_{\mathbf{x}\in\mathbb{R}^d} \sum_{i=1}^{n} \|\mathbf{x} - \mathbf{x}_i\|. \quad (2)$$

In other words, the geometric median is the vector that minimizes the sum of $L_2$-distances. When the vectors $\mathbf{x}_1,\ldots,\mathbf{x}_n$ are not collinear, the optimization problem in the right-hand side of (2) has a unique solution (Vardit & Zhang, 2000).

The values of $\epsilon$ for the three $(f,\kappa)$-robust aggregators are summarized in Table 2. Due to the limited space, the analysis details for TM, CM and GM are deferred to Appendix A.3, Appendix A.4 and Appendix A.5, respectively. Moreover, please note that the value of $\epsilon$ for the coordinate-wise trimmed mean $TM_{f/n}$ exactly meets the general lower bound in Theorem 3.1. It indicates that the value of $\epsilon$ for $TM_{f/n}$ and the general lower bound are both tight.

### 3.4. Conditions of the Worst Case

We have obtained that the value of $\epsilon$ can be up to $\frac{f}{n-f}$ in the worst case. Then we would like to focus on the following question:

*Under what conditions is the aggregation error largest?*

To answer the question above, we would like to first present the theoretical result and a proof sketch for the aggregation error of TM in Theorem 3.6 below. The conditions of the worst case can be derived based on the proof of Theorem 3.6.

**Theorem 3.6.** *When $f < \frac{n}{2}$, the coordinate-wise trimmed mean* $TM_{f/n}$ *is* $\frac{f}{n-f}$-*accurate.*

#### 3.4.1. PROOF SKETCH OF THEOREM 3.6

We first consider the case where the dimension $d = 1$. Let $x_1,\ldots,x_n$ be any $n$ real numbers. Without loss of generality, we assume that $x_1 \leq x_2 \leq \ldots \leq x_n$ since the order does not affect the trimmed mean of the $n$ values. Furthermore, we let sets $I_- = \{1,2,\ldots,f\}$, $I_0 = \{f+1,\ldots,n-f\}$ and $I_+ = \{n-f+1,\ldots,n\}$, and define

$$\bar{x}_{I_-} = \frac{1}{|I_-|}\sum_{i\in I_-} x_i = \frac{1}{f}\sum_{i=1}^{f} x_i, \quad (3)$$

$$\bar{x}_{I_0} = \frac{1}{|I_0|}\sum_{i\in I_0} x_i = \frac{1}{n-2f}\sum_{i=f+1}^{n-f} x_i, \quad (4)$$

$$\bar{x}_{I_+} = \frac{1}{|I_+|}\sum_{i\in I_+} x_i = \frac{1}{f}\sum_{i=n-f+1}^{n} x_i. \quad (5)$$

According to the definition,

$$\bar{x} = \frac{f}{n}\bar{x}_{I_-} + \frac{n-2f}{n}\bar{x}_{I_0} + \frac{f}{n}\bar{x}_{I_+},$$

$$\text{TM}_{f/n}(x_1,\ldots,x_n) = \bar{x}_{I_0}.$$

Thus, we have

$$\text{TM}_{f/n}(x_1,\ldots,x_n) - \bar{x}$$
$$= \frac{f}{n}(\bar{x}_{I_0} - \bar{x}_{I_-}) + \frac{f}{n}(\bar{x}_{I_0} - \bar{x}_{I_+}) = \frac{f}{n}\Delta_- - \frac{f}{n}\Delta_+, \quad (6)$$

where

$$\Delta_- = \bar{x}_{I_0} - \bar{x}_{I_-} \geq 0, \quad \text{and} \quad \Delta_+ = \bar{x}_{I_+} - \bar{x}_{I_0} \geq 0.$$

Moreover, we define the within-group variances

$$s_{I_-}^2 = \frac{1}{f}\sum_{i\in I_-}(x_i - \bar{x}_{I_-})^2, \qquad s_{I_0}^2 = \frac{1}{f}\sum_{i\in I_0}(x_i - \bar{x}_{I_0})^2,$$

$$s_{I_+}^2 = \frac{1}{f}\sum_{i\in I_+}(x_i - \bar{x}_{I_+})^2,$$

and we have the following equation

$$
\frac{1}{n} \sum_{i=1}^{n} (x_i - \bar{x})^2 = \frac{n-f}{f} \left[ \mathrm{TM}_{f/n}(x_1, \ldots, x_n) - \bar{x} \right]^2
$$
$$
+ \frac{2f}{n}(\Delta_- \Delta_+) + \left[ \frac{f}{n} \cdot s_{I_-}^2 + \frac{n-2f}{n} \cdot s_{I_0}^2 + \frac{f}{n} \cdot s_{I_+}^2 \right].
$$
(7)

For space saving, the proof details of (7) are deferred to Appendix A.3. Noticing that $s_{I_-}^2 \geq 0$, $s_{I_0}^2 \geq 0$, $s_{I_+}^2 \geq 0$, $\Delta_- \geq 0$ and $\Delta_+ \geq 0$, we have that $\frac{1}{n} \sum_{i=1}^{n} (x_i - \bar{x})^2 \geq \frac{n-f}{f} \left[ \mathrm{TM}_{f/n}(x_1, \ldots, x_n) - \bar{x} \right]^2$, or, equivalently,

$$
\left[ \mathrm{TM}_{f/n}(x_1, \ldots, x_n) - \bar{x} \right]^2 \leq \frac{f}{n-f} \cdot \left[ \frac{1}{n} \sum_{i=1}^{n} (x_i - \bar{x})^2 \right].
$$
(8)

For general $d$-dimension cases, $\|\mathrm{TM}_{f/n}(\mathbf{x}_1, \ldots, \mathbf{x}_n) - \bar{\mathbf{x}}\|^2$ and $\|\mathbf{x}_i - \bar{\mathbf{x}}\|^2$ can be written as summations of square errors on each dimension. Recursively using inequality (8) for 1-dimension case above, we finally obtain that

$$
\|\mathrm{TM}_{f/n}(\mathbf{x}_1, \ldots, \mathbf{x}_n) - \bar{\mathbf{x}}\|^2 \leq \frac{f}{n-f} \cdot \left[ \frac{1}{n} \sum_{i=1}^{n} \|\mathbf{x}_i - \bar{\mathbf{x}}\|^2 \right].
$$
(9)

It implies that $\mathrm{TM}_{f/n}$ is $\frac{f}{n-f}$-accurate.

### 3.4.2. DISCUSSION ON THE WORST CASE

By comparing Equation (7) and Inequality (8), we can find that the equation in (8) holds if and only if

$$
\frac{2f}{n}(\Delta_- \Delta_+) + \left[ \frac{f}{n} \cdot s_{I_-}^2 + \frac{n-2f}{n} \cdot s_{I_0}^2 + \frac{f}{n} \cdot s_{I_+}^2 \right] = 0,
$$

which is equivalent to that $s_{I_-}^2 = s_{I_0}^2 = s_{I_+}^2 = 0$ and $\Delta_- \Delta_+ = 0$. Without loss of generality, we assume that $\Delta_+ = 0$. Recall that $s_{I_-}^2$, $s_{I_0}^2$ and $s_{I_+}^2$ are the within-group variances of $\{x_i\}_{i \in I_-}$, $\{x_i\}_{i \in I_0}$ and $\{x_i\}_{i \in I_+}$, respectively. Therefore, we have

$$
x_1 = x_2 = \ldots = x_f \quad \text{and} \quad x_{f+1} = \ldots = x_n. \quad (10)
$$

In summary, for 1-dimension case, in the case described by Equation (10), the aggregation error is the largest. For the general $d$-dimension case, the aggregation error is the largest when Equation (10) holds for each dimension.

Moreover, please note that the condition described by Equation (10) indicates a large skewness among $\{x_1, \ldots, x_n\}$. Thus, our theoretical results show that a large data skewness will lead to a large error of robust aggregation even if there is actually no Byzantine workers.

### 3.5. More Discussion

#### 3.5.1. ABOUT THE TIGHTNESS OF THE BOUNDS

As we can see, the value of $\epsilon$ for $\mathrm{TM}_{f/n}$ exactly meets the lower bound $\frac{f}{n-f}$. Thus, both the upper bound for $\mathrm{TM}_{f/n}$ and the lower bound are tight. Moreover, since coordinate-wise median (CM) is robust against $f = \lfloor \frac{n-1}{2} \rfloor$ Byzantine workers, the value of $\epsilon$ for CM also meets the lower bound.

Then we show the tightness of the value of $\epsilon$ for GM. Consider the case where $n = 2n'$ is an even number and the dimension $d = 1$. Moreover, $\mathbf{x}_1 = \mathbf{x}_2 = \ldots = \mathbf{x}_{n'} = 0$ and $\mathbf{x}_{n'+1} = \mathbf{x}_{n'+2} = \ldots = \mathbf{x}_{2n'} = 1$. In this case, $\bar{\mathbf{x}} = \frac{1}{2}$ and $\frac{1}{n} \sum_{i=1}^{n} \|\bar{\mathbf{x}} - \mathbf{x}_i\|^2 = \frac{1}{n} \left( \sum_{i=1}^{n} \frac{1}{4} \right) = \frac{1}{4}$. The geometric median $\mathbf{x}_{GM}$ can be any value in $[0, 1]$. When $\mathbf{x}_{GM} = 0$ or 1, we have that $\|\mathbf{x}_{GM} - \bar{\mathbf{x}}\|^2 = \frac{1}{4}$. Therefore, in this case,

$$
\|\mathbf{x}_{GM} - \bar{\mathbf{x}}\|^2 = \frac{1}{4} = 1 \cdot \left[ \frac{1}{n} \sum_{i=1}^{n} \|\bar{\mathbf{x}} - \mathbf{x}_i\|^2 \right].
$$

Therefore, the value $\epsilon = 1$ for GM in Table 2 is tight. Moreover, the value of $\epsilon$ for GM asymptotically meets the lower bound since GM is robust against $f = \lfloor \frac{n-1}{2} \rfloor$ Byzantine workers and $\lim_{n \to \infty} \lfloor \frac{n-1}{2} \rfloor / (n - \lfloor \frac{n-1}{2} \rfloor) = 1$.

#### 3.5.2. ABOUT DATA HETEROGENEITY

Due to the tightness of the lower bound, the aggregation error in the worst case is proportional to $\frac{1}{n} \sum_{i=1}^{n} \|\mathbf{x}_i - \bar{\mathbf{x}}\|^2$, which could be large when the probability distributions of $\mathbf{x}_i$'s are heterogeneous. Therefore, the aggregation error could be larger when data distribution are more heterogeneous. There are some existing techniques to reduce the gradient (or momentum) heterogeneity in BRDL such as bucketing (Karimireddy et al., 2022) and nearest neighbour mixing (NNM) (Allouah et al., 2023). However, both of the two techniques requires the prior knowledge of the maximum number of Byzantine workers to mix the input vectors. Less Byzantine workers can be tolerated if we want to make the vectors more homogeneous after mixing.

From another perspective, an aggregator combined with NNM (or bucketing) can be considered as a new aggregator that can resist fewer Byzantine workers but have less aggregation error. Therefore, when using NNM or bucketing, we actually use the prior knowledge of the Byzantine worker number to make a better trade-off between robustness and accuracy. Please see Section 5 for the empirical results, which will further support the theoretical findings in this section.

#### 3.5.3. SUMMARY OF THEORETICAL RESULTS

In this section, we theoretically show that for any $(f, \kappa)$-robust aggregator, when there are no Byzantine workers, it is inevitably to have an aggregation error in the order of

$O(\frac{f}{n-f})$ in the worst case. Moreover, the value of $\frac{f}{n-f}$ increases as $f$ increases. It shows that to obtain robustness against more Byzantine workers, the worst-case performance in no-attack cases is inevitably degraded, which we deem as the tension between Byzantine robustness and no-attack accuracy.

## 4. Convergence

In this section, we theoretically analyze the convergence of ByzGD with $(f, \kappa)$-robust aggregators when there are actually no Byzantine workers. Firstly, we list the assumptions below.

**Assumption 4.1** (Bounded loss). $\exists F^* \in \mathbb{R}$ such that $\forall \mathbf{w} \in \mathbb{R}^d, F(\mathbf{w}) \geq F^*.$

**Assumption 4.2** (*L*-smoothness). $\forall \mathbf{w}, \mathbf{w}' \in \mathbb{R}^d$, $\|\nabla F(\mathbf{w}) - \nabla F(\mathbf{w}')\| \leq L\|\mathbf{w} - \mathbf{w}'\|.$

**Assumption 4.3** (Bounded heterogeneity). There exists $G \geq 0$ such that $\forall \mathbf{w} \in \mathbb{R}^d,$

$$\frac{1}{n} \sum_{i=1}^{n} \|\nabla F_i(\mathbf{w}) - \nabla F(\mathbf{w})\|^2 \leq G^2.$$

**Assumption 4.4** (Polyak-Łojasiewicz (PL) condition). Let $F^*$ denote the lower bound of $F(\mathbf{w})$. $\exists \mu > 0$ such that $\forall \mathbf{w} \in \mathbb{R}^d, F(\mathbf{w}) - F^* \leq \frac{1}{2\mu}\|\nabla F(\mathbf{w})\|^2.$

The four assumptions above are common in distributed learning (Allouah et al., 2023; Blanchard et al., 2017; Farhadkhani et al., 2022; Karimireddy et al., 2021; 2022; Xie et al., 2019; 2020b). Specifically, the value of $G$ measures the heterogeneity of local gradients on workers. A larger $G$ typically makes the distributed learning task more challenging. Under the assumptions, we have the following convergence result for ByzGD.

**Theorem 4.5** (Lower bound). *For ByzGD (Algorithm 1) with any $(f, \kappa)$-robust aggregator, there exist $n$ loss functions $F_1(\mathbf{w}), \ldots, F_n(\mathbf{w})$ satisfying Assumptions 4.1, 4.2, 4.3 and 4.4, and satisfying the following condition: For any initial point $\mathbf{w}_0$, any positive constant learning rate $\eta_t = \eta > 0$, any constant $C_1 < \frac{f}{n-f}G^2$ and any positive integer $K > 0$, there exists an integer $T > K$ such that*

$$\frac{1}{T} \sum_{t=0}^{T-1} \|\nabla F(\mathbf{w}_t)\|^2 > C_1,$$

*and*

$$F(\mathbf{w}_T) - F^* > \frac{C_1}{2\mu}.$$

Due to the limited space, we only provide a proof sketch of Theorem 4.5 here. Consider the 1-dimension case where

$$F_1(w) = \ldots = F_f(w) = \frac{nG}{2\sqrt{f(n-f)}}w^2$$

and

$$F_{f+1}(w) = \ldots = F_n(w) = \frac{nG}{2\sqrt{f(n-f)}}(w^2 - 2w).$$

It can be verified that $F(w) = \frac{1}{n}\sum_{i=1}^{n} F_i(w)$ has the unique global minimum point $w^* = \frac{n-f}{n}$ and satisfies Assumptions 4.1, 4.2, 4.3 and 4.4. Moreover, the updating rule of ByzGD for this case can be written as:

$$w_{t+1} - 1 = \left[1 - \frac{nG\eta}{\sqrt{f(n-f)}}\right](w_t - 1).$$

By separately analyzing the three cases where (i) $0 < \eta < \frac{2\sqrt{f(n-f)}}{nG}$; (ii) $\eta = \frac{2\sqrt{f(n-f)}}{nG}$; and (iii) $\eta > \frac{2\sqrt{f(n-f)}}{nG}$, we can finally obtain the desired results. Please refer to Appendix A for more proof details.

Theorem 4.5 indicates that whatever the initial point $\mathbf{w}_0$, the learning rate $\eta$ and the iteration number $T$ are, ByzGD with a $(f, \kappa)$-robust aggregator cannot guarantee $\frac{1}{T}\sum_{t=0}^{T-1}\|\nabla F(\mathbf{w}_t)\|^2$ to be smaller than $\frac{f}{n-f}G^2$ for general non-convex loss functions that satisfy Assumptions 4.1, 4.2 and 4.3. Similarly, $[F(\mathbf{w}_T) - F^*]$ cannot be guaranteed to be smaller than $\frac{f}{n-f} \cdot \frac{G^2}{2\mu}$ for loss functions that satisfy Assumptions 4.1, 4.2, 4.3 and 4.4.

In general cases where $\nabla F_i(\mathbf{w})$'s are not all the same, we have $G > 0$. Theorem 4.5 indicates that ByzGD with a $(f, \kappa)$-robust aggregator cannot guarantee the convergence to 0. Moreover, the term $\frac{f}{n-f}$ increases with the increase of $f$, where $f$ is the number of Byzantine workers that can be tolerated. Therefore, the convergence lower bound will be larger if more Byzantine workers can be tolerated, which we consider as the tension between Byzantine robustness and no-attack accuracy. Then we show that the constant terms $\frac{f}{n-f}G^2$ and $\frac{f}{n-f} \cdot \frac{G^2}{2\mu}$ are optimal and cannot be improved.

**Theorem 4.6** (Upper bound). *For ByzGD (Algorithm 1), under Assumption 4.1, Assumption 4.2 and Assumption 4.3, when $\eta_t = \frac{1}{L}$ and $\mathbf{Agg}(\cdot)$ is $\epsilon$-accurate, we have that*

$$\frac{1}{T} \sum_{t=0}^{T-1} \|\nabla F(\mathbf{w}_t)\|^2 \leq \frac{2L[F^* - F(\mathbf{w}_T)]}{T} + \epsilon G^2.$$

*Furthermore, when $F(\mathbf{w})$ also satisfies the PL condition (Assumption 4.4), we have that*

$$F(\mathbf{w}_T) - F^* \leq \left(1 - \frac{\mu}{L}\right)^T [F(\mathbf{w}_0) - F^*] + \frac{\epsilon G^2}{2\mu}.$$

The terms $\frac{2L[F^* - F(\mathbf{w}_T)]}{T}$ and $(1 - \frac{\mu}{L})^T[F(\mathbf{w}_0) - F^*]$ in Theorem 4.6 approach 0 when $T \to \infty$, while the constant terms $\epsilon G^2$ and $\frac{\epsilon G^2}{2\mu}$ remain. Please note that $\epsilon \geq \frac{f}{n-f}$ (Theorem 3.1) and that $TM_{f/n}$ is both $(f, \kappa)$-robust and $\epsilon$-accurate

with $\epsilon = \frac{f}{n-f}$. It indicates that the lower bound $\frac{f}{n-f}G^2$ for $C_1$ in Theorem 4.5 cannot be improved. Therefore, the constant terms in Theorem 4.5 and those in Theorem 4.6 are both tight.

In this section, we mainly analyze the convergence of ByzGD. In some large-scale problems, the computation of local gradients is time-consuming. Some variants such as Byzantine-robust stochastic gradient descent (ByzSGD) has a much lower computation cost of each iteration and is more widely used. We would like to point out that the lower bound in Theorem 4.5 can be directly applied to ByzSGD since ByzGD can be considered as a special case of ByzSGD with a zero variance. Local momentum (Allouah et al., 2023; Farhadkhani et al., 2022; Karimireddy et al., 2021) is also proposed to reduce the variance of stochastic gradients in ByzSGD. However, the constant terms $\epsilon G^2$ and $\frac{\epsilon G^2}{2\mu}$ are related to the aggregation accuracy $\epsilon$ and the heterogeneity degree $G$ while using stochastic gradients or local momentums does not help to reduce $\epsilon$ or $G$. Stochastic gradients can reduce the computation cost of each iteration, but do not have a better worst-case convergence guarantee. Therefore, for these variants, there are similar tensions between Byzantine robustness and no-attack accuracy, which we will verify by empirical results.

In real-world applications, the number of Byzantine workers is usually unknown, and it is typically required to determine in advance the number $f$ of Byzantine workers that the distributed learning system can tolerate. A too small $f$ will make the BRDL method easy to be foiled by Byzantine workers. However, a too large $f$ may also lead to a large aggregation error, and thus degrade the model accuracy, even if there are actually no Byzantine workers. Thus, $f$ should be carefully estimated before designing the distributed learning system in real-world applications.

## 5. Experiment

In this section, we will empirically test the effect of using robust aggregator when there are no Byzantine workers. Specifically, we use ByzSGD with various robust aggregators to train a ResNet-20 (He et al., 2016) deep learning model on the CIFAR-10 dataset (Krizhevsky et al., 2009) for 160 epochs without attacks. All the experiments are conducted on a distributed platform with 16 Docker containers serving as workers and an extra Docker container as the server. Each Docker container is bound to an NVIDIA TITAN Xp GPU. We test the performance of each method when the training instances are randomly distributed to the workers according to the Dirichlet distribution with hyper-parameters $\alpha = 0.1, 1.0$ and $10.0$, respectively. A smaller $\alpha$ will lead to a more heterogeneous data distribution. Moreover, the batch normalization (BN) layers in the ResNet-20 model are replaced with group normalization layers since

BN layers have a poor performance with heterogeneous data across workers (Wu & He, 2018). All algorithms are implemented with PyTorch 1.3.

We use cross-entropy as the loss function, set the batch size on each worker to 16, and use the cosine annealing learning rates (Loshchilov & Hutter, 2017). Specifically, the learning rate at the $p$-th epoch is $\eta_p = \frac{1+\cos(p\pi/160)}{2}\eta_0$ for $p = 0, 1, \ldots, 159$. The initial learning rate $\eta_0$ is selected from $\{0.1, 0.2, 0.5, 1.0\}$, and the best final top-1 test accuracy is used as the final metrics. Local momentum is used with momentum hyper-parameter set to 0.9. We first test the performance of using multi-Krum (Blanchard et al., 2017) and coordinate-wise trimmed mean (Yin et al., 2018) when hyper-parameter $f$, which is the number of the Byzantine workers that can be tolerated, ranges from 0 to 7. It takes about 1.5 hours to run each method for 160 epochs.

As the results in Table 3 and Table 4 show, for both multi-Krum and coordinate-wise trimmed mean, the final top-1 test accuracy decreases as $f$ increases. In other words, to make the BRDL method robust to more Byzantine workers, the model accuracy under no attack will decrease. Furthermore, the final top-1 test accuracy decreases more rapidly with a smaller $\alpha$. Please note that $\alpha$ is the hyper-parameter related to the data distribution. A smaller $\alpha$ indicates a more heterogeneous data distribution and a larger $G$ (please refer to Assumption 4.3). Thus, the final top-1 test accuracy decreases more rapidly when $G$ is larger, which is consistent with the theoretical results in Theorem 4.5 and Theorem 4.6.

We also test the performance of multi-Krum and coordinate-wise trimmed mean with nearest neighbour mixing (NNM) (Allouah et al., 2023). The NNM technique can reduce the heterogeneity of vectors. Specifically, we fix $f = 7$ (the maximum number of Byzantine workers that can be tolerated) for both of the two robust aggregators and let the hyper-parameter $f_{\text{NNM}}$ range from 1 to 7 for NNM. A smaller $f_{\text{NNM}}$ makes the vectors after mixing more homogeneous but will also decrease the number of Byzantine workers that the whole BRDL method can tolerate. Specifically, the whole BRDL method is robust to $\min\{f, f_{\text{NNM}}\}$ Byzantine workers. When $f_{\text{NNM}} = 0$, the output vectors of NNM will be all the same and equal the mean of the input vectors. Thus, when $f_{\text{NNM}} = 0$, robust aggregator combined with NNM is equivalent to vanilla mean aggregator.

As the results in Table 5 and Table 6 show, using NNM alleviates the degrade of final top-1 test accuracy for both multi-Krum and coordinate-wise trimmed mean. Moreover, using a smaller $f_{\text{NNM}}$ can lead to a higher final top-1 test accuracy, but will also make the distributed learning method robust to less Byzantine workers. Therefore, there is also a tension between Byzantine robustness and no-attack accuracy when using the NNM technique.

*Table 3.* The final top-1 test accuracy of using multi-Krum with various hyper-parameter $f$ to train the model for 160 epochs under different data distributions when there are no Byzantine workers. Both of the two aggregators are equivalent to vanilla mean when $f = 0$. The values in the parentheses are the differences compared to the case of $f = 0$.

| $f$ | Multi-Krum (Blanchard et al., 2017) | | |
|---|---|---|---|
| | $\alpha = 0.1$ | $\alpha = 1.0$ | $\alpha = 10.0$ |
| 0 (=mean) | 89.42% | 89.36% | 89.55% |
| 1 | 88.05% (-1.37%) | 87.50% (-1.86%) | 88.36% (-1.19 %) |
| 3 | 83.50% (-5.92%) | 84.49% (-4.87%) | 87.02% (-2.53%) |
| 5 | 69.86% (-19.56%) | 80.40% (-8.96%) | 84.64% (-4.91%) |
| 7 | 40.31% (-49.11%) | 68.54% (-20.82%) | 73.69% (-15.86%) |

*Table 4.* The final top-1 test accuracy of using coordinate-wise trimmed mean with various hyper-parameter $f$ to train the model for 160 epochs under different data distributions when there are no Byzantine workers. Both of the two aggregators are equivalent to vanilla mean when $f = 0$. The values in the parentheses are the differences compared to the case of $f = 0$.

| $f$ | Coordinate-wise trimmed mean (Yin et al., 2018) | | |
|---|---|---|---|
| | $\alpha = 0.1$ | $\alpha = 1.0$ | $\alpha = 10.0$ |
| 0 (=mean) | 89.42% | 89.36% | 89.55% |
| 1 | 50.13% (-39.29%) | 66.18% (-23.18%) | 88.71% (-0.84%) |
| 3 | 24.95% (-64.47%) | 48.32% (-41.04%) | 87.51% (-2.04%) |
| 5 | 10.00% (-79.42%) | 42.86% (-46.50%) | 86.68% (-2.87%) |
| 7 | 10.00% (-79.42%) | 40.54% (-48.82%) | 83.15% (-6.40%) |

*Table 5.* The final top-1 test accuracy of using multi-Krum with NNM to train the model for 160 epochs under different data distributions when there are no Byzantine workers. Both of the two robust aggregators combined with NNM are equivalent to vanilla mean when $f_{\text{NNM}} = 0$. The values in the parentheses are the differences compared to the case of $f_{\text{NNM}} = 0$.

| $f_{\text{NNM}}$ | Multi-Krum (Blanchard et al., 2017) with $f = 7$ | | |
|---|---|---|---|
| | $\alpha = 0.1$ | $\alpha = 1.0$ | $\alpha = 10.0$ |
| 0 (=mean) | 89.42% | 89.36% | 89.55% |
| 1 | 88.81% (-0.61%) | 88.39% (-0.97%) | 89.49% (-0.06%) |
| 3 | 87.61% (-1.81%) | 86.56% (-2.80%) | 88.93% (-0.62%) |
| 5 | 83.16% (-6.26%) | 84.63% (-4.73%) | 87.99% (-1.56%) |
| 7 | 75.03% (-14.39%) | 83.88% (-5.48%) | 86.94% (-2.61%) |
| without NNM | 40.31% (-49.11%) | 68.54% (-20.82%) | 73.69% (-15.86%) |

*Table 6.* The final top-1 test accuracy of using coordinate-wise trimmed mean with NNM to train the model for 160 epochs under different data distributions when there are no Byzantine workers. Both of the two robust aggregators combined with NNM are equivalent to vanilla mean when $f_{\text{NNM}} = 0$. The values in the parentheses are the differences compared to the case of $f_{\text{NNM}} = 0$.

| $f_{\text{NNM}}$ | Coordinate-wise trimmed mean (Yin et al., 2018) with $f = 7$ | | |
|---|---|---|---|
| | $\alpha = 0.1$ | $\alpha = 1.0$ | $\alpha = 10.0$ |
| 0 (=mean) | 89.42% | 89.36% | 89.55% |
| 1 | 86.61% (-2.81%) | 88.59% (-0.77%) | 88.57% (-0.98%) |
| 3 | 83.83% (-5.59%) | 87.57% (-1.79%) | 88.53% (-1.02%) |
| 5 | 81.50% (-7.92%) | 86.03% (-3.33%) | 88.06% (-1.49%) |
| 7 | 61.09% (-28.33%) | 82.77% (-6.59%) | 87.79% (-1.76%) |
| without NNM | 10.00% (-79.42%) | 40.54% (-48.82%) | 83.15% (-6.40%) |

# 6. Conclusion

To the best of our knowledge, this is the first work that theoretically investigates the tension between Byzantine robustness and no-attack accuracy in distributed learning. We theoretically analyze the aggregation error of robust aggregators and the convergence rate of using gradient descent with robust aggregators. The theoretical results show that making the distributed learning method robust to more Byzantine workers will degrade the worst-case performance under no attack. Our theoretical findings are further supported by the empirical results.

# Acknowledgements

This work is supported by National Key R&D Program of China (No.2020YFA0713900), NSFC Project (No.12326615), the Key Major Project of the Pengcheng Laboratory (No.PCL2024A06).

# Impact Statement

This paper mainly theoretically explores the tension between Byzantine robustness and no-attack accuracy in distributed learning. There may be some potential societal consequences of our work, none of which we feel must be specifically highlighted here.

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

# A. Proof Details

## A.1. Proof of Theorem 3.1

*Proof.* Let $\mathbf{Agg}(\cdot)$ be a $(f, \kappa)$-robust aggregator. Assume that $\mathbf{Agg}(\cdot)$ is $\epsilon$-accurate. We consider the case where $\mathbf{x}_1 = \ldots = \mathbf{x}_{n-f} = 0$ and $\mathbf{x}_{n-f+1} = \ldots = \mathbf{x}_n = A > 0$. We have

$$\bar{\mathbf{x}} = \frac{1}{n} \sum_{i=1}^{n} \mathbf{x}_i = \frac{f}{n} \cdot A, \tag{11}$$

and

$$\frac{1}{n} \sum_{i=1}^{n} \|\mathbf{x}_i - \bar{\mathbf{x}}\|^2 = \frac{1}{n} \left[ (n-f) \cdot \left( \frac{f}{n} \cdot A \right)^2 + f \cdot \left( A - \frac{f}{n} \cdot A \right)^2 \right] = \frac{f(n-f)}{n^2} \cdot A^2. \tag{12}$$

Let set $S' = \{1, \ldots, n-f\}$. We have

$$\bar{\mathbf{x}}_{S'} = \frac{1}{|S'|} \sum_{i \in S'} \mathbf{x}_i = 0,$$

and

$$\frac{1}{|S'|} \sum_{i \in S'} \|\mathbf{x}_i - \bar{\mathbf{x}}_{S'}\|^2 = 0.$$

According to the definition of $(f, \kappa)$-robustness, we have

$$\|\mathbf{Agg}(\mathbf{x}_1, \ldots, \mathbf{x}_n) - \bar{\mathbf{x}}_{S'}\|^2 \le \kappa \cdot \left[ \frac{1}{|S'|} \sum_{i \in S'} \|\mathbf{x}_i - \bar{\mathbf{x}}_{S'}\|^2 \right] = 0,$$

which implies that

$$\mathbf{Agg}(\mathbf{x}_1, \ldots, \mathbf{x}_n) = \bar{\mathbf{x}}_{S'} = \mathbf{0}. \tag{13}$$

According to equations (11), (12) and (13),

$$\|\mathbf{Agg}(\mathbf{x}_1, \ldots, \mathbf{x}_n) - \bar{\mathbf{x}}\|^2 = \left( \mathbf{0} - \frac{f}{n} \cdot A \right)^2 = \frac{f^2}{n^2} \cdot A^2 = \frac{f}{n-f} \cdot \left[ \frac{1}{n} \sum_{i=1}^{n} \|\mathbf{x}_i - \bar{\mathbf{x}}\|^2 \right].$$

In addition, according to the definition of $\epsilon$-accuracy,

$$\|\mathbf{Agg}(\mathbf{x}_1, \ldots, \mathbf{x}_n) - \bar{\mathbf{x}}\|^2 \le \epsilon \cdot \left[ \frac{1}{n} \sum_{i=1}^{n} \|\mathbf{x}_i - \bar{\mathbf{x}}\|^2 \right].$$

Therefore,

$$\epsilon \cdot \left[ \frac{1}{n} \sum_{i=1}^{n} \|\mathbf{x}_i - \bar{\mathbf{x}}\|^2 \right] \ge \|\mathbf{Agg}(\mathbf{x}_1, \ldots, \mathbf{x}_n) - \bar{\mathbf{x}}\|^2 = \frac{f}{n-f} \cdot \left[ \frac{1}{n} \sum_{i=1}^{n} \|\mathbf{x}_i - \bar{\mathbf{x}}\|^2 \right].$$

Consequently, it is obtained that

$$\epsilon \ge \frac{f}{n-f}.$$

$\square$

## A.2. Proof of Theorem 3.2

*Proof.* Suppose that $\mathbf{Agg}(\cdot)$ is an $(f, \kappa)$-robust aggregator. Let set

$$\mathcal{U} = \{S | S \subseteq \{1, \ldots, n\}, |S| = n - f\}.$$

Therefore, we have

$$|\mathcal{U}| = \binom{n}{n-f} = \frac{n!}{f!(n-f)!}.$$

For any $S \in \mathcal{U}$ and any $n$ vectors $\mathbf{x}_1, \ldots, \mathbf{x}_n \in \mathbb{R}^d$, we have

$$
\begin{aligned}
\sum_{i \in S} \|\mathbf{x}_i - \bar{\mathbf{x}}_S\|^2 &= \sum_{i \in S} \left( \|\mathbf{x}_i\|^2 - 2\mathbf{x}_i \cdot \bar{\mathbf{x}}_S + \|\bar{\mathbf{x}}_S\|^2 \right) \\
&= \sum_{i \in S} \|\mathbf{x}_i\|^2 - 2|S|\bar{\mathbf{x}}_S \cdot \bar{\mathbf{x}}_S + |S|\|\bar{\mathbf{x}}_S\|^2 \\
&= \sum_{i \in S} \|\mathbf{x}_i\|^2 - |S|\|\bar{\mathbf{x}}_S\|^2 \\
&= \sum_{i \in S} \|\mathbf{x}_i\|^2 - \frac{1}{|S|} \left\| \sum_{i \in S} \mathbf{x}_i \right\|^2 \\
&= \sum_{i \in S} \|\mathbf{x}_i\|^2 - \frac{1}{|S|} \sum_{i \in S} \|\mathbf{x}_i\|^2 - \frac{2}{|S|} \sum_{i \in S, j \in S, i < j} \mathbf{x}_i \cdot \mathbf{x}_j
\end{aligned}
$$

Notice that $|S| = n - f$ for all $S \in \mathcal{U}$. Therefore,

$$
\sum_{i \in S} \|\mathbf{x}_i - \bar{\mathbf{x}}_S\|^2 = \frac{n - f - 1}{n - f} \sum_{i \in S} \|\mathbf{x}_i\|^2 - \frac{2}{n - f} \sum_{i, j \in S, i < j} \mathbf{x}_i \cdot \mathbf{x}_j. \tag{14}
$$

Take summation over all $S \in \mathcal{U}$ for both sides of (14), it is obtained that

$$
\sum_{S \in \mathcal{U}} \left( \sum_{i \in S} \|\mathbf{x}_i - \bar{\mathbf{x}}_S\|^2 \right) = \frac{n - f - 1}{n - f} \sum_{S \in \mathcal{U}} \left( \sum_{i \in S} \|\mathbf{x}_i\|^2 \right) - \frac{2}{n - f} \sum_{S \in \mathcal{U}} \left( \sum_{i, j \in S, i < j} \mathbf{x}_i \cdot \mathbf{x}_j \right).
$$

For any element $i \in \{1, \ldots, n\}$, there are totally $\binom{n-1}{n-f-1}$ sets $S$ in $\mathcal{U}$ that contain $i$. Moreover, for any two different elements $i, j \in \{1, \ldots, n\}$, there are totally $\binom{n-2}{n-f-2}$ sets $S$ in $\mathcal{U}$ that contain both $i$ and $j$. Therefore,

$$
\begin{aligned}
&\sum_{S \in \mathcal{U}} \left( \sum_{i \in S} \|\mathbf{x}_i - \bar{\mathbf{x}}_S\|^2 \right) \\
&= \frac{n - f - 1}{n - f} \binom{n - 1}{n - f - 1} \sum_{i=1}^{n} \|\mathbf{x}_i\|^2 - \frac{2}{n - f} \binom{n - 2}{n - f - 2} \sum_{i=1}^{n-1} \sum_{j=i+1}^{n} \mathbf{x}_i \cdot \mathbf{x}_j \\
&= \frac{(n - f - 1)(n - 1)!}{(n - f)f!(n - f - 1)!} \sum_{i=1}^{n} \|\mathbf{x}_i\|^2 - \frac{2(n - 2)!}{(n - f)f!(n - f - 2)!} \sum_{i=1}^{n-1} \sum_{j=i+1}^{n} \mathbf{x}_i \cdot \mathbf{x}_j \\
&= \frac{(n - f - 1)(n - 1)!}{(n - f)!f!} \sum_{i=1}^{n} \|\mathbf{x}_i\|^2 - \frac{2(n - 2)!(n - f - 1)}{(n - f)!f!} \sum_{i=1}^{n-1} \sum_{j=i+1}^{n} \mathbf{x}_i \cdot \mathbf{x}_j \\
&= \frac{n!}{f!(n - f)!} \cdot \frac{n - f - 1}{n - 1} \left( \frac{n - 1}{n} \sum_{i=1}^{n} \|\mathbf{x}_i\|^2 - \frac{2}{n} \sum_{i=1}^{n-1} \sum_{j=i+1}^{n} \mathbf{x}_i \cdot \mathbf{x}_j \right).
\end{aligned}
$$

In (14), let $S = \{1, \ldots, n\}$, and we have

$$
\sum_{i=1}^{n} \|\mathbf{x}_i - \bar{\mathbf{x}}\|^2 = \frac{n - 1}{n} \sum_{i=1}^{n} \|\mathbf{x}_i\|^2 - \frac{2}{n} \sum_{i=1}^{n-1} \sum_{j=i+1}^{n} \mathbf{x}_i \cdot \mathbf{x}_j, \tag{15}
$$

where $\bar{\mathbf{x}} = \frac{1}{n} \sum_{i=1}^{n} \mathbf{x}_i$. Therefore,

$$
\sum_{S \in \mathcal{U}} \left( \sum_{i \in S} \|\mathbf{x}_i - \bar{\mathbf{x}}_S\|^2 \right) = \frac{n!}{f!(n - f)!} \cdot \frac{n - f - 1}{n - 1} \sum_{i=1}^{n} \|\mathbf{x}_i - \bar{\mathbf{x}}\|^2.
$$

Since $|\mathcal{U}| = \frac{n!}{f!(n-f)!}$ and $|S| = n - f$ for all $S \in \mathcal{U}$, it is obtained that

$$\frac{1}{|\mathcal{U}|} \sum_{S \in \mathcal{U}} \left( \frac{1}{|S|} \sum_{i \in S} \|\mathbf{x}_i - \bar{\mathbf{x}}_S\|^2 \right) = \frac{(n-f-1)n}{(n-f)(n-1)} \left( \frac{1}{n} \sum_{i=1}^{n} \|\mathbf{x}_i - \bar{\mathbf{x}}\|^2 \right)$$

$$= \frac{n^2 - nf - n}{n^2 - nf - n + f} \left( \frac{1}{n} \sum_{i=1}^{n} \|\mathbf{x}_i - \bar{\mathbf{x}}\|^2 \right).$$

Therefore,

$$\frac{1}{|\mathcal{U}|} \sum_{S \in \mathcal{U}} \left( \frac{1}{|S|} \sum_{i \in S} \|\mathbf{x}_i - \bar{\mathbf{x}}_S\|^2 \right) \leq \frac{1}{n} \sum_{i=1}^{n} \|\mathbf{x}_i - \bar{\mathbf{x}}\|^2. \tag{16}$$

Meanwhile, by using Cauchy's inequality and the fact that $\bar{\mathbf{x}} = \frac{1}{|\mathcal{U}|} \sum_{S \in \mathcal{U}} \bar{\mathbf{x}}_S$, we have

$$\|\mathbf{Agg}(\mathbf{x}_1, \ldots, \mathbf{x}_n) - \bar{\mathbf{x}}\|^2 = \left\| \frac{1}{|\mathcal{U}|} \sum_{S \in \mathcal{U}} [\mathbf{Agg}(\mathbf{x}_1, \ldots, \mathbf{x}_n) - \bar{\mathbf{x}}_S] \right\|^2$$

$$\leq \frac{1}{|\mathcal{U}|} \sum_{S \in \mathcal{U}} \|\mathbf{Agg}(\mathbf{x}_1, \ldots, \mathbf{x}_n) - \bar{\mathbf{x}}_S\|^2.$$

According to the definition of $(f, \kappa)$-robustness,

$$\|\mathbf{Agg}(\mathbf{x}_1, \ldots, \mathbf{x}_n) - \bar{\mathbf{x}}_S\|^2 \leq \kappa \cdot \left[ \frac{1}{|S|} \sum_{i \in S} \|\mathbf{x}_i - \bar{\mathbf{x}}_S\|^2 \right],$$

Therefore,

$$\|\mathbf{Agg}(\mathbf{x}_1, \ldots, \mathbf{x}_n) - \bar{\mathbf{x}}\|^2 \leq \kappa \cdot \frac{1}{|\mathcal{U}|} \sum_{S \in \mathcal{U}} \left( \frac{1}{|S|} \sum_{i \in S} \|\mathbf{x}_i - \bar{\mathbf{x}}_S\|^2 \right).$$

Using inequality (16), we finally obtained that

$$\|\mathbf{Agg}(\mathbf{x}_1, \ldots, \mathbf{x}_n) - \bar{\mathbf{x}}\|^2 \leq \kappa \cdot \left[ \frac{1}{n} \sum_{i=1}^{n} \|\mathbf{x}_i - \bar{\mathbf{x}}\|^2 \right], \tag{17}$$

which implies that $\mathbf{Agg}(\cdot)$ is $\kappa$-accurate. $\qquad \square$

### A.3. Analysis for Coordinate-wise Trimmed Mean

*Proof.* We first proof the theorem for the case where the dimension $d = 1$. Let $x_1, \ldots, x_n$ be any $n$ real numbers. Without loss of generality, we assume that $x_1 \leq x_2 \leq \ldots \leq x_n$ since the order does not affect the trimmed mean of the $n$ values, and the trimmed mean can be written as

$$\mathrm{TM}_{f/n}(x_1, \ldots, x_n) = \frac{1}{n - 2f} \sum_{i=f+1}^{n-f} x_i.$$

Therefore,

$$\mathrm{TM}_{f/n}(x_1, \ldots, x_n) - \bar{x}$$

$$= \frac{1}{n - 2f} \sum_{i=f+1}^{n-f} x_i - \frac{1}{n} \sum_{i=1}^{n} x_i$$

$$= \left( \frac{1}{n - 2f} - \frac{1}{n} \right) \left[ \sum_{i=f+1}^{n-f} x_i \right] - \frac{1}{n} \left[ \sum_{i=1}^{f} x_i \right] - \frac{1}{n} \left[ \sum_{i=n-f+1}^{n} x_i \right]$$

$$= \frac{2f}{n} \left[ \frac{1}{n-2f} \sum_{i=f+1}^{n-f} x_i \right] - \frac{f}{n} \left[ \frac{1}{f} \sum_{i=1}^{f} x_i \right] - \frac{f}{n} \left[ \frac{1}{f} \sum_{i=n-f+1}^{n} x_i \right]$$

$$= \frac{f}{n} \left[ \frac{1}{n-2f} \sum_{i=f+1}^{n-f} x_i - \frac{1}{f} \sum_{i=1}^{f} x_i \right] + \frac{f}{n} \left[ \frac{1}{n-2f} \sum_{i=f+1}^{n-f} x_i - \frac{1}{f} \sum_{i=n-f+1}^{n} x_i \right].$$

Let sets $I_- = \{1, 2, \ldots, f\}$, $I_0 = \{f+1, \ldots, n-f\}$ and $I_+ = \{n-f+1, \ldots, n\}$. We define

$$\bar{x}_{I_-} = \frac{1}{|I_-|} \sum_{i \in I_-} x_i = \frac{1}{f} \sum_{i=1}^{f} x_i,$$

$$\bar{x}_{I_0} = \frac{1}{|I_0|} \sum_{i \in I_0} x_i = \frac{1}{n-2f} \sum_{i=f+1}^{n-f} x_i,$$

and

$$\bar{x}_{I_+} = \frac{1}{|I_+|} \sum_{i \in I_+} x_i = \frac{1}{f} \sum_{i=n-f+1}^{n} x_i.$$

Therefore, we have

$$\text{TM}_{f/n}(x_1, \ldots, x_n) - \bar{x} = \frac{f}{n}(\bar{x}_{I_0} - \bar{x}_{I_-}) + \frac{f}{n}(\bar{x}_{I_0} - \bar{x}_{I_+}). \tag{18}$$

Meanwhile,

$$\frac{1}{n} \sum_{i=1}^{n} (x_i - \bar{x})^2 = \frac{1}{n} \sum_{i \in I_-} (x_i - \bar{x})^2 + \frac{1}{n} \sum_{i \in I_0} (x_i - \bar{x})^2 + \frac{1}{n} \sum_{i \in I_+} (x_i - \bar{x})^2$$

$$= \frac{1}{n} \sum_{i \in I_-} [(x_i - \bar{x}_{I_-}) + (\bar{x}_{I_-} - \bar{x})]^2$$

$$+ \frac{1}{n} \sum_{i \in I_0} [(x_i - \bar{x}_{I_0}) + (\bar{x}_{I_0} - \bar{x})]^2$$

$$+ \frac{1}{n} \sum_{i \in I_+} [(x_i - \bar{x}_{I_+}) + (\bar{x}_{I_+} - \bar{x})]^2. \tag{19}$$

Notice that

$$\frac{1}{n} \sum_{i \in I_-} [(x_i - \bar{x}_{I_-}) + (\bar{x}_{I_-} - \bar{x})]^2$$

$$= \frac{1}{n} \sum_{i \in I_-} [(x_i - \bar{x}_{I_-})^2 + (\bar{x}_{I_-} - \bar{x})^2 + 2(x_i - \bar{x}_{I_-})(\bar{x}_{I_-} - \bar{x})]$$

$$= \frac{1}{n} \sum_{i \in I_-} (x_i - \bar{x}_{I_-})^2 + \frac{1}{n} \sum_{i \in I_-} (\bar{x}_{I_-} - \bar{x})^2 + \frac{1}{n} \sum_{i \in I_-} [2(x_i - \bar{x}_{I_-})(\bar{x}_{I_-} - \bar{x})]$$

$$= \frac{f}{n} \left[ \frac{1}{f} \sum_{i \in I_-} (x_i - \bar{x}_{I_-})^2 \right] + \frac{f}{n}(\bar{x}_{I_-} - \bar{x})^2 + \frac{2(\bar{x}_{I_-} - \bar{x})}{n} \sum_{i \in I_-} (x_i - \bar{x}_{I_-}).$$

Since $\sum_{i \in I_-} (x_i - \bar{x}_{I_-}) = 0$, we have

$$\frac{1}{n} \sum_{i \in I_-} [(x_i - \bar{x}_{I_-}) + (\bar{x}_{I_-} - \bar{x})]^2 = \frac{f}{n} \cdot [s_{I_-}^2 + (\bar{x}_{I_-} - \bar{x})^2], \tag{20}$$

where $s_{I_-}^2 = \frac{1}{f} \sum_{i \in I_-} (x_i - \bar{x}_{I_-})^2$. Similarly,

$$\frac{1}{n} \sum_{i \in I_0} [(x_i - \bar{x}_{I_0}) + (\bar{x}_{I_0} - \bar{x})]^2 = \frac{n - 2f}{n} \cdot [s_{I_0}^2 + (\bar{x}_{I_0} - \bar{x})^2], \tag{21}$$

$$\frac{1}{n} \sum_{i \in I_+} [(x_i - \bar{x}_{I_+}) + (\bar{x}_{I_+} - \bar{x})]^2 = \frac{f}{n} \cdot [s_{I_+}^2 + (\bar{x}_{I_+} - \bar{x})^2], \tag{22}$$

where $s_{I_0}^2 = \frac{1}{n-2f} \sum_{i \in I_0} (x_i - \bar{x}_{I_0})^2$ and $s_{I_+}^2 = \frac{1}{f} \sum_{i \in I_+} (x_i - \bar{x}_{I_+})^2$. Substituting (20), (21) and (22) into (19), we have

$$\frac{1}{n} \sum_{i=1}^{n} (x_i - \bar{x})^2 = \left[ \frac{f}{n} \cdot s_{I_-}^2 + \frac{n - 2f}{n} \cdot s_{I_0}^2 + \frac{f}{n} \cdot s_{I_+}^2 \right]$$

$$+ \frac{f}{n} \cdot (\bar{x}_{I_-} - \bar{x})^2 + \frac{n - 2f}{n} \cdot (\bar{x}_{I_0} - \bar{x})^2 + \frac{f}{n} \cdot (\bar{x}_{I_+} - \bar{x})^2.$$

Notice that $\bar{x} = \frac{f}{n} \bar{x}_{I_-} + \frac{n-2f}{n} \bar{x}_{I_0} + \frac{f}{n} \bar{x}_{I_+}$, and we have

$$\frac{1}{n} \sum_{i=1}^{n} (x_i - \bar{x})^2 = \left[ \frac{f}{n} \cdot s_{I_-}^2 + \frac{n - 2f}{n} \cdot s_{I_0}^2 + \frac{f}{n} \cdot s_{I_+}^2 \right]$$

$$+ \frac{f}{n} \cdot \left[ \frac{n - 2f}{n} (\bar{x}_{I_-} - \bar{x}_{I_0}) + \frac{f}{n} (\bar{x}_{I_-} - \bar{x}_{I_+}) \right]^2$$

$$+ \frac{n - 2f}{n} \cdot \left[ \frac{f}{n} (\bar{x}_{I_0} - \bar{x}_{I_-}) + \frac{f}{n} (\bar{x}_{I_0} - \bar{x}_{I_+}) \right]^2$$

$$+ \frac{f}{n} \cdot \left[ \frac{f}{n} (\bar{x}_{I_+} - \bar{x}_{I_-}) + \frac{n - 2f}{n} (\bar{x}_{I_+} - \bar{x}_{I_0}) \right]^2.$$

For simplicity, we let

$$\tilde{s}^2 = \frac{f}{n} \cdot s_{I_-}^2 + \frac{n - 2f}{n} \cdot s_{I_0}^2 + \frac{f}{n} \cdot s_{I_+}^2 \geq 0,$$

$$\Delta_- = \bar{x}_{I_0} - \bar{x}_{I_-} \geq 0, \quad \text{and} \quad \Delta_+ = \bar{x}_{I_+} - \bar{x}_{I_0} \geq 0.$$

Therefore, we have

$$\frac{1}{n} \sum_{i=1}^{n} (x_i - \bar{x})^2 = \tilde{s}^2 + \frac{f}{n} \cdot \left[ \frac{n - 2f}{n} (-\Delta_-) + \frac{f}{n} (-\Delta_- - \Delta_+) \right]^2$$

$$+ \frac{n - 2f}{n} \cdot \left[ \frac{f}{n} (\Delta_-) + \frac{f}{n} (-\Delta_+) \right]^2$$

$$+ \frac{f}{n} \cdot \left[ \frac{f}{n} (\Delta_- + \Delta_+) + \frac{n - 2f}{n} (\Delta_+) \right]^2.$$

By expanding the square terms, it is obtained that

$$\frac{1}{n} \sum_{i=1}^{n} (x_i - \bar{x})^2$$

$$= \frac{f}{n} \left( \frac{n - f}{n} \Delta_- + \frac{f}{n} \Delta_+ \right)^2 + \frac{n - 2f}{n} \left( \frac{f}{n} \Delta_- - \frac{f}{n} \Delta_+ \right)^2 + \frac{f}{n} \left( \frac{f}{n} \Delta_- + \frac{n - f}{n} \Delta_+ \right)^2 + \tilde{s}^2$$

$$= \left( \frac{f(n - f)^2}{n^3} + \frac{f^2(n - 2f)}{n^3} + \frac{f^3}{n^3} \right) (\Delta_-)^2 + \left( \frac{f^3}{n^3} + \frac{f^2(n - 2f)}{n^3} + \frac{f(n - f)^2}{n^3} \right) (\Delta_+)^2$$

$$+ \left( \frac{2f^2(n - f)}{n^3} - \frac{2f^2(n - 2f)}{n^3} + \frac{2f^2(n - f)}{n^3} \right) (\Delta_- \Delta_+) + \tilde{s}^2$$

$$= \frac{f(n-f)}{n^2}(\Delta_-)^2 + \frac{f(n-f)}{n^2}(\Delta_+)^2 + \frac{2f^2}{n^2}(\Delta_-\Delta_+) + \tilde{s}^2$$

$$= \frac{n-f}{f}\left(\frac{f}{n}\Delta_- - \frac{f}{n}\Delta_+\right)^2 + \frac{2f}{n}(\Delta_-\Delta_+) + \tilde{s}^2.$$

Meanwhile, according to (18) and the definition of $\Delta_-$ and $\Delta_+$, we have

$$\frac{f}{n}\Delta_- - \frac{f}{n}\Delta_+ = \text{TM}_{f/n}(x_1, \ldots, x_n) - \bar{x}.$$

Therefore,

$$\frac{1}{n}\sum_{i=1}^{n}(x_i - \bar{x})^2 = \frac{n-f}{f}\left[\text{TM}_{f/n}(x_1, \ldots, x_n) - \bar{x}\right]^2 + \frac{2f}{n}(\Delta_-\Delta_+) + \tilde{s}^2.$$

Notice that $\tilde{s}^2 \geq 0, \Delta_- \geq 0$ and $\Delta_+ \geq 0$, and we finally obtain that

$$\frac{1}{n}\sum_{i=1}^{n}(x_i - \bar{x})^2 \geq \frac{n-f}{f}\left[\text{TM}_{f/n}(x_1, \ldots, x_n) - \bar{x}\right]^2,$$

or, equivalently,

$$\left[\text{TM}_{f/n}(x_1, \ldots, x_n) - \bar{x}\right]^2 \leq \frac{f}{n-f}\cdot\left[\frac{1}{n}\sum_{i=1}^{n}(x_i - \bar{x})^2\right]. \tag{23}$$

Moreover, the equality in (23) holds if and only if $\tilde{s}^2 = 0$ and $\Delta_-\Delta_+ = 0$, which is equivalent to that $s_{I_-}^2 = s_{I_0}^2 = s_{I_+}^2 = 0$ and $\Delta_-\Delta_+ = 0$.

We have finished the proof for the 1-dimension case. For the general $d$-dimension case,

$$\|\text{TM}_{f/n}(\mathbf{x}_1, \ldots, \mathbf{x}_n) - \bar{\mathbf{x}}\|^2 = \sum_{j=1}^{d}\left([\text{TM}_{f/n}(\mathbf{x}_1, \ldots, \mathbf{x}_n) - \bar{\mathbf{x}}]_j\right)^2$$

$$= \sum_{j=1}^{d}\left([\text{TM}_{f/n}(\mathbf{x}_1, \ldots, \mathbf{x}_n)]_j - [\bar{\mathbf{x}}]_j\right)^2$$

$$= \sum_{j=1}^{d}\left(\text{TM}_{f/n}([\mathbf{x}_1]_j, \ldots, [\mathbf{x}_n]_j) - \frac{1}{n}\sum_{i=1}^{n}[\mathbf{x}_i]_j\right)^2$$

$$\overset{\text{(by (23))}}{\leq} \sum_{j=1}^{d}\left(\frac{f}{n-f}\cdot\frac{1}{n}\sum_{i=1}^{n}([\mathbf{x}_i]_j - [\bar{\mathbf{x}}]_j)^2\right)$$

$$= \sum_{j=1}^{d}\left(\frac{f}{n-f}\cdot\frac{1}{n}\sum_{i=1}^{n}[\mathbf{x}_i - \bar{\mathbf{x}}]_j^2\right)$$

$$= \frac{f}{n-f}\cdot\frac{1}{n}\sum_{i=1}^{n}\|\mathbf{x}_i - \bar{\mathbf{x}}\|^2.$$

Namely,

$$\|\text{TM}_{f/n}(\mathbf{x}_1, \ldots, \mathbf{x}_n) - \bar{\mathbf{x}}\|^2 \leq \frac{f}{n-f}\cdot\left[\frac{1}{n}\sum_{i=1}^{n}\|\mathbf{x}_i - \bar{\mathbf{x}}\|^2\right].$$

$\square$

## A.4. Analysis for Coordinate-wise Median

**Theorem A.1.** *Coordinate-wise median is $\epsilon_n$-accurate where $\epsilon_n = \frac{\lfloor\frac{n-1}{2}\rfloor}{n-\lfloor\frac{n-1}{2}\rfloor}$.*

*Proof.* Notice that coordinate-wise median can be deemed as $\frac{\lfloor\frac{n-1}{2}\rfloor}{n}$-TM. The desired result can be directly obtained by applying Theorem 3.6. $\square$

## A.5. Analysis for Geometric Median

**Theorem A.2.** *The geometric median is* 1-*accurate.*

*Proof.* We use $\mathbf{x}_{GM}$ to denote the geometric median of $\mathbf{x}_1, \ldots, \mathbf{x}_n$. According to the definition, we have the following inequality:

$$\sum_{i=1}^{n} \|\mathbf{x}_{GM} - \mathbf{x}_i\| \leq \sum_{i=1}^{n} \|\bar{\mathbf{x}} - \mathbf{x}_i\|. \tag{24}$$

According to the Triangle Inequality,

$$\|\mathbf{x}_{GM} - \bar{\mathbf{x}}\|^2 = \left\| \mathbf{x}_{GM} - \frac{1}{n} \sum_{i=1}^{n} \mathbf{x}_i \right\|^2 = \left\| \frac{1}{n} \sum_{i=1}^{n} (\mathbf{x}_{GM} - \mathbf{x}_i) \right\|^2 \leq \left[ \frac{1}{n} \sum_{i=1}^{n} \|\mathbf{x}_{GM} - \mathbf{x}_i\| \right]^2. \tag{25}$$

Combining (24) and (25), we have

$$\|\mathbf{x}_{GM} - \bar{\mathbf{x}}\|^2 \leq \left[ \frac{1}{n} \sum_{i=1}^{n} \|\bar{\mathbf{x}} - \mathbf{x}_i\| \right]^2 = \frac{1}{n^2} \left[ \sum_{i=1}^{n} \|\bar{\mathbf{x}} - \mathbf{x}_i\| \right]^2. \tag{26}$$

According to Cauchy-Schwarz inequality, we have

$$\left( \sum_{i=1}^{n} 1^2 \right) \cdot \left( \sum_{i=1}^{n} \|\bar{\mathbf{x}} - \mathbf{x}_i\|^2 \right) \geq \left( \sum_{i=1}^{n} 1 \cdot \|\bar{\mathbf{x}} - \mathbf{x}_i\| \right)^2,$$

Namely,

$$\left[ \sum_{i=1}^{n} \|\bar{\mathbf{x}} - \mathbf{x}_i\| \right]^2 \leq n \cdot \sum_{i=1}^{n} \|\bar{\mathbf{x}} - \mathbf{x}_i\|^2. \tag{27}$$

Combining (26) and (27), we finally obtain that

$$\|\mathbf{x}_{GM} - \bar{\mathbf{x}}\|^2 \leq \frac{1}{n^2} \left[ n \cdot \sum_{i=1}^{n} \|\bar{\mathbf{x}} - \mathbf{x}_i\|^2 \right] = 1 \cdot \left[ \frac{1}{n} \sum_{i=1}^{n} \|\bar{\mathbf{x}} - \mathbf{x}_i\|^2 \right].$$

It implies that geometric median is 1-accurate. $\qquad \square$

## A.6. Proof of Theorem 4.5

*Proof.* We first consider the 1-dimension case where

$$F_1(w) = \ldots = F_f(w) = \frac{nG}{2\sqrt{f(n-f)}} w^2$$

and

$$F_{f+1}(w) = \ldots = F_n(w) = \frac{nG}{2\sqrt{f(n-f)}} w^2 - \frac{nG}{\sqrt{f(n-f)}} w.$$

Therefore,

$$
\begin{aligned}
F(w) = \frac{1}{n} \sum_{i=1}^{n} F_i(w) &= \frac{f}{n} \times \frac{nG}{2\sqrt{f(n-f)}} w^2 + \frac{n-f}{n} \times \left[ \frac{nG}{2\sqrt{f(n-f)}} w^2 - \frac{nG}{\sqrt{f(n-f)}} w \right] \\
&= \frac{nG}{2\sqrt{f(n-f)}} w^2 - \frac{\sqrt{n-f}}{\sqrt{f}} Gw \\
&= \frac{nG}{2\sqrt{f(n-f)}} \left( w - \frac{n-f}{n} \right)^2 - \frac{(n-f)^{\frac{3}{2}} G}{2n\sqrt{f}},
\end{aligned}
$$

which indicates that $F(w)$ satisfies Assumptions 4.1 and 4.2 and has the unique global mimimum point

$$w^* = \frac{n - f}{n}.$$

Moreover,

$$
\begin{aligned}
\frac{1}{n} \sum_{i=1}^{n} [F_i'(w) - F'(w)]^2 &= \frac{f}{n} \times \left( \frac{\sqrt{n-f}}{\sqrt{f}} G \right)^2 + \frac{n-f}{n} \times \left( \frac{nG}{\sqrt{f(n-f)}} - \frac{\sqrt{n-f}}{\sqrt{f}} G \right)^2 \\
&= \frac{n-f}{n} G^2 + \frac{f}{n} G^2 \\
&= G^2.
\end{aligned}
$$

Therefore, $F(w)$ satisfies Assumptions 4.3. Furthermore, $F(w)$ also satisfies Assumption 4.4 with $\mu = \frac{nG}{\sqrt{f(n-f)}}$ since $F(w)$ is a quadratic function. Let the set $S = \{f + 1, \ldots, n\}$ and

$$F_S'(w) = \frac{1}{|S|} \sum_{i \in S} F_i'(w) = \frac{nG}{\sqrt{f(n-f)}} (w - 1).$$

Since $\mathbf{Agg}(\cdot)$ is $(f, \kappa)$-robust and $F_{f+1}'(w) = \ldots = F_n'(w)$, we have

$$[\mathbf{Agg}(F_1'(w), \ldots, F_n'(w)) - F_S'(w)]^2 \le \kappa \cdot \left[ \frac{1}{|S|} \sum_{i \in S} (F_i'(w) - F_S'(w))^2 \right] = 0.$$

Therefore, we have

$$\mathbf{Agg}(F_1'(w), \ldots, F_n'(w)) = \frac{nG}{\sqrt{f(n-f)}} (w - 1).$$

Let $\{w_t\}_{t=0}^{\infty}$ be the sequence obtained by ByzGD (Algorithm 1) with initial point $w_0 \in \mathbb{R}$ and learning rate $\eta_t = \eta > 0$. We have

$$w_{t+1} = w_t - \eta \cdot \mathbf{Agg}(F_1'(w_t), \ldots, F_n'(w_t)) = w_t - \frac{nG\eta}{\sqrt{f(n-f)}} (w_t - 1).$$

Therefore,

$$w_{t+1} - 1 = \left[ 1 - \frac{nG\eta}{\sqrt{f(n-f)}} \right] (w_t - 1).$$

Then we consider the following three cases.

- Case (i). When $0 < \eta < \frac{2\sqrt{f(n-f)}}{nG}$, we have $1 - \frac{nG\eta}{\sqrt{f(n-f)}} \in (-1, 1)$.

Thus, $w_t - 1 = \left[ 1 - \frac{nG\eta}{\sqrt{f(n-f)}} \right]^t (w_0 - 1)$ converges to 0. Consequently, $w_t$ converges to 1. Since $F(w)$ is $L$-smooth, $[F'(w_t)]^2$ converges to $[F'(1)]^2 = \left[ \frac{nG}{\sqrt{f(n-f)}} (1 - \frac{n-f}{n}) \right]^2 = \frac{f}{n-f} G^2$. Thus, for any $C_1 < \frac{f}{n-f} G^2$ and any positive $K$, there exists $T_1 > K$ such that

$$\frac{1}{T_1} \sum_{t=0}^{T_1 - 1} [F'(w_t)]^2 > C_1.$$

Similarly, we have that $[F(w_t) - F(w^*)]$ converges to

$$F(1) - F(w^*) = \frac{nG}{2\sqrt{f(n-f)}} \left( 1 - \frac{n-f}{n} \right)^2 = \frac{f^2 G}{2n\sqrt{f(n-f)}} = \frac{f}{n-f} \cdot \frac{G^2}{2\mu}$$

by noticing that $\mu = \frac{nG}{\sqrt{f(n-f)}}$. Thus, for any $C_1 < \frac{f}{n-f}G^2$ and any positive $K$, there exists $T_2 > K$ such that

$$F(w_{T_2}) - F(w^*) > \frac{C_1}{2\mu}.$$

Let $T = \max\{T_1, T_2\}$ and we obtain the desired result.

- Case (ii). When $\eta = \frac{2\sqrt{f(n-f)}}{nG}$, we have $1 - \frac{nG\eta}{\sqrt{f(n-f)}} = -1$. Thus, $w_{t+1} - 1 = -(w_t - 1)$. For any positive $K$, we have that $w_{\lfloor K \rfloor + 2} - 1 = -(w_{\lfloor K \rfloor + 2} - 1)$, which is equivalent to that $w_{\lfloor K \rfloor + 1} + w_{\lfloor K \rfloor + 2} = 2$. Let

$$T_1 = \underset{t \in \{\lfloor K \rfloor + 1, \lfloor K \rfloor + 2\}}{\arg\max} \; w_t.$$

We have that $w_{T_1} = \max\{w_{\lfloor K \rfloor + 1}, w_{\lfloor K \rfloor + 2}\} \geq 1$. Since $F(w)$ is monotonically increasing when $w \geq 1$, we have that

$$F(w_{T_1}) - F(w^*) \geq F(1) - F(w^*) = \frac{f}{n-f} \cdot \frac{G^2}{2\mu} > \frac{C_1}{2\mu}.$$

Let $H = \frac{1}{T_1} \sum_{t=0}^{T_1 - 1} [F'(w_t)]^2$. If $H > C_1$, we directly obtain the desired result by letting $T = T_1$. If $H \leq C_1$, let

$$T = T_1 + 2 \left\lfloor \frac{(n-f)(TC_1 - T_1 H)}{2fG^2} + 1 \right\rfloor,$$

where $\lfloor \cdot \rfloor$ is the floor function. Therefore,

$$w_T - 1 = (w_{T_1} - 1) \times (-1)^{2 \left\lfloor \frac{(n-f)(TC_1 - T_1 H)}{2fG^2} + 1 \right\rfloor} = w_{T_1} - 1.$$

Namely, $w_T = w_{T_1}$. Therefore,

$$F(w_T) - F(w^*) = F(w_{T_1}) - F(w^*) > \frac{C_1}{2\mu}.$$

Moreover, using the equation that $w_t + w_{t+1} = 2$ and the inequality that $x^2 + y^2 \geq \frac{(x+y)^2}{2}$,

$$[F'(w_t)]^2 + [F'(w_{t+1})]^2$$

$$= \left[ \frac{nG}{\sqrt{f(n-f)}} \left( w_t - \frac{n-f}{n} \right) \right]^2 + \left[ \frac{nG}{\sqrt{f(n-f)}} \left( w_{t+1} - \frac{n-f}{n} \right) \right]^2$$

$$= \frac{n^2 G^2}{f(n-f)} \left[ \left( w_t - \frac{n-f}{n} \right)^2 + \left( w_{t+1} - \frac{n-f}{n} \right)^2 \right]$$

$$\geq \frac{n^2 G^2}{2f(n-f)} \left( w_t + w_{t+1} - \frac{2(n-f)}{n} \right)^2$$

$$= \frac{2n^2 G^2}{f(n-f)} \left( 1 - \frac{n-f}{n} \right)^2$$

$$= \frac{2f}{n-f} G^2.$$

Thus,

$$\frac{1}{T} \sum_{t=0}^{T-1} [F'(w_t)]^2 = \frac{1}{T} \sum_{t=0}^{T_1 - 1} [F'(w_t)]^2 + \frac{1}{T} \sum_{t=T_1}^{T-1} [F'(w_t)]^2$$

$$= \frac{T_1}{T} \cdot H + \frac{1}{T} \cdot \frac{T - T_1}{2} \cdot \frac{2f}{n-f} G^2$$

$$= \frac{T_1 H}{T} + \frac{1}{T} \cdot \left\lfloor \frac{(n-f)(TC_1 - T_1 H)}{2fG^2} + 1 \right\rfloor \frac{2f}{n-f} G^2$$

$$> \frac{T_1 H}{T} + \frac{1}{T} \cdot \frac{(n-f)(TC_1 - T_1 H)}{2fG^2} \cdot \frac{2f}{n-f} G^2$$

$$= \frac{T_1 H}{T} + \frac{TC_1 - T_1 H}{T}$$

$$= C_1.$$

Therefore, the positive integer $T \geq T_1 > K$ satisfies the two desired conditions.

- Case (iii). When $\eta > \frac{2\sqrt{f(n-f)}}{nG}$, we have $1 - \frac{nG\eta}{\sqrt{f(n-f)}} < -1$. In this case, $w_t \to \infty$. Since $F(w)$ is $L$-smooth, and both $[F'(w)]^2$ and $F(w)$ diverge to $+\infty$ when $w \to \infty$. We can directly obtain the desired result.

In summary, the desired result can be obtained in all of the three cases, and the proof for the 1-dimension case is finished.

For general $d$-dimension cases where $\mathbf{w} \in \mathbb{R}^d$, we can simply choose $F_i(\mathbf{w}) = \tilde{F}_i(\mathbf{w}_{[1]})$ for all $i \in \{1, 2, \dots, n\}$ where $\mathbf{w}_{[1]}$ is the first coordinate of vector $\mathbf{w}$. Thus, we have that $\|\nabla F(\mathbf{w})\|^2 = [\tilde{F}'(\mathbf{w}_{[1]})]^2$. Therefore, we can use the result for the 1-dimension case above, and obtain the desired result for general $d$-dimension cases. □

### A.7. Proof of Theorem 4.6

*Proof.* Let $\mathbf{G}_t = \mathbf{Agg}(\nabla F_1(\mathbf{w}_t), \dots, \nabla F_n(\mathbf{w}_t))$. Since $F(\mathbf{w})$ is $L$-smooth,

$$F(\mathbf{w}_{t+1}) = F(\mathbf{w}_t - \eta_t \mathbf{G}_t)$$

$$\leq F(\mathbf{w}_t) - \eta_t \nabla F(\mathbf{w}_t)^T \mathbf{G}_t + \frac{L(\eta_t)^2}{2} \|\mathbf{G}_t\|^2$$

$$= F(\mathbf{w}_t) - \frac{\eta_t}{2} \|\nabla F(\mathbf{w}_t)\|^2 + \frac{\eta_t}{2} \|\mathbf{G}_t - \nabla F(\mathbf{w}_t)\|^2 - \frac{\eta_t(1 - L\eta_t)}{2} \|\mathbf{G}_t\|^2, \tag{28}$$

where the last equation can be quickly checked by expanding the squares. Moreover, using Assumption 4.3 and that $\mathbf{Agg}(\cdot)$ is $\epsilon$-accurate, we have that

$$\|\mathbf{G}_t - \nabla F(\mathbf{w}_t)\|^2 \leq \epsilon \cdot \left[ \frac{1}{n} \sum_{i=1}^{n} \|\nabla F_i(\mathbf{w}) - \nabla F(\mathbf{w})\|^2 \right] \leq \epsilon G^2. \tag{29}$$

Since $\eta_t = \frac{1}{L}$, we have that $1 - L\eta_t = 0$. Substituting (29) into (28), we have that

$$F(\mathbf{w}_{t+1}) - F(\mathbf{w}_t) \leq -\frac{\eta_t}{2} \|\nabla F(\mathbf{w}_t)\|^2 + \frac{\eta_t}{2} \epsilon G^2 = -\frac{1}{2L} \|\nabla F(\mathbf{w}_t)\|^2 + \frac{1}{2L} \epsilon G^2. \tag{30}$$

Taking summation over $t$, it is obtained that

$$\frac{1}{T} \sum_{t=0}^{T-1} \|\nabla F(\mathbf{w}_t)\|^2 \leq \frac{2L[F(\mathbf{w}_0) - F(\mathbf{w}_T)]}{T} + \epsilon G^2.$$

According to Assumption 4.1, we have that $F(\mathbf{w}_T) \geq F^*$ and obtain the first desired result. Furthermore, when $F(\mathbf{w})$ satisfies the PL condition, we have that

$$\|\nabla F(\mathbf{w}_t)\|^2 \geq 2\mu[F(\mathbf{w}_t) - F^*]. \tag{31}$$

Substituting (31) into (30), we have that

$$F(\mathbf{w}_{t+1}) - F(\mathbf{w}_t) \leq -\frac{\mu}{L}[F(\mathbf{w}_t) - F^*] + \frac{\epsilon G^2}{2L},$$

which is equivalent to that

$$\left[ F(\mathbf{w}_{t+1}) - F^* - \frac{\epsilon G^2}{2\mu} \right] \leq \left(1 - \frac{\mu}{L}\right) \left[ F(\mathbf{w}_t) - F^* - \frac{\epsilon G^2}{2\mu} \right].$$

Recursively using the inequality, we have that

$$F(\mathbf{w}_T) - F^* - \frac{\epsilon G^2}{2\mu} \leq \left(1 - \frac{\mu}{L}\right)^T \left[F(\mathbf{w}_0) - F^* - \frac{\epsilon G^2}{2\mu}\right].$$

Therefore, we finally obtain that

$$F(\mathbf{w}_T) - F^* \leq \left(1 - \frac{\mu}{L}\right)^T \left[F(\mathbf{w}_0) - F^*\right] + \frac{\epsilon G^2}{2\mu}.$$

$\square$

