# OpenReview forum: "On the Tension between Byzantine Robustness and No-Attack Accuracy in Distributed Learning"
_ICML.cc/2025/Conference — ICML 2025 spotlightposter_

### Official Review · Reviewer_x4mZ · 2025-03-08

**Overall Recommendation:** 3

**Summary:**

This paper explores the trade-off between Byzantine robustness and standard accuracy in distributed learning. It provides a theoretical analysis of the error of robust aggregation methods when there are no Byzantine workers.  In doing so, it establishes lower bounds on the deviation from the average of the datapoints as well as on the convergence rate of Byzantine robust gradient descent (ByzGD) in the absence of Byzantine workers. The authors present theoretical results demonstrating that the worst-case aggregation error (with respect to the average) increases as the number of expected Byzantine workers increases, leading to a potential degradation in accuracy. Empirical experiments are provided to support the theoretical findings.

**Claims And Evidence:**

The paper claims that making an aggregator more robust to Byzantine workers leads to an inevitable degradation in standard accuracy when there are no Byzantine failures. To support this claim, the paper studies the notion of **worst-case accuracy** of a robust aggregator $\textbf{Agg}$, defined as

$$ \epsilon := \sup_{x_1, \dots, x_n \in \mathbb{R}^d}   \frac{\Vert\textbf{Agg}(x_1, \dots, x_n) - \bar{x}_n  \Vert^2}{\frac{1}{n}  \Sigma \Vert x_i - \bar{x}_n \Vert^2} . $$

$\textbf{Note:}$ I am rewriting the definition here (off course excluding the trivial case of zero empirical variance).

Theoretical results support the claim by proving lower bounds on a well-known class of robust aggregation methods known as $(f,\kappa)$-robust averaging introduced in [1]. This bound writes $\epsilon \in \Omega(\frac{f}{n-f})$, hence it linearly depends on the ratio $\frac{f}{n-f}$ where $n$ is the total number workers in the system and $f$ is the maximal amount of Byzantine workers the aggregation can theoretically tolerate. Based on this idea, the paper also provides a lower bound on the wort-case training error of ByzGD. These lower bounds are tight as proven by the matching upper bounds the paper presents.

Empirical experiments show accuracy degradation for ByzGD in the absence of Byzantine workers in several context to further support the point made by the theoretical claims (this point will be mitigated below).

**Essential References Not Discussed:**

There are some recent works that are very related to the Byzantine literature like [3,4] that could be cited but are not essential to the understanding of the paper.

[3] Robust Distributed Learning: Tight Error Bounds and Breakdown Point under Data Heterogeneity, Youssef Allouah Rachid Guerraoui Nirupam Gupta Rafaël Pinot Geovani Rizk, (Neurips 2023)
[4] Variance reduction is an antidote to Byzantoine workers, Eduard Gorbunov, Samuel Horvath, Peter Richtarik, Gauthier Gidel (ICLR 2023)

**Experimental Designs Or Analyses:**

The experimental setup seemed sound to me.

**Methods And Evaluation Criteria:**

The authors define worst-case accuracy using the norm difference from the true average of the datapoints (see above). The worst-case scenario (i.e., the lower bound) is constructed by assuming datasets with extreme skewness, which significantly affects the distance between the computed aggregation and the true average of the distribution. However, in such cases, targeting the average as the aggregation goal may not be statistically meaningful anymore (as the average is not a meaningful summary of the dataset anymore). Accordingly, I am not certain to understand the semantic of being distant from the average in such a scenario. In essence, I am unsure that the worst-case analysis is the best way to capture the accuracy vs robustness trade-off of Byzantine robust methods. A more informative approach would be to analyze statistical bounds under assumed data distributions to better model realistic data heterogeneity among honest workers.

I also feel like there is a lack of deep analysis on state-of-the-art robust aggregation techniques. In fact, the paper does not truly discusses the impact of state-of-the-art pre-aggregation methods like NNM or Bucketing on the trade-off being studied. These methods have been shown to improve Byzantine robustness while mitigating accuracy degradation, making their omission a significant gap in my opinion. I also feel similar concerns on the experimental part. While the experiments confirm theoretical findings on simple aggregation rules like trimmed mean or median, they do not seem to explain very clearly the behavior of state-of-the-art methods like NNM or Bucketing.

**Side note:** I would suggest including loss curves in the empirical evaluation to study whether the theoretical findings hold throughout the training process (as the theory in on training not testing).

**Other Comments Or Suggestions:**

This paper raises an interesting and important question about the trade-off between Byzantine robustness and accuracy in non-attack scenarios. However, I am unsure if it currently provides a deep enough analysis of this trade-off (I will wait for discussion with the authors and the other reviewers to make my final decision I guess). Here are some suggestions, that might help Improve the paper.

- Consider statistical bounds under realistic data distributions rather than focusing solely on worst-case skewed data.
- Expand the analysis of state-of-the-art robust aggregation techniques like NNM and Bucketing.
- Clarify the novelty of the proof techniques and how they differ from prior work.
- Strengthen the empirical evaluation by including loss curves and more in-depth analysis of heterogeneity effects.

**Other Strengths And Weaknesses:**

NA

**Questions For Authors:**

Please answer my above concerns, especially regarding proof novelty and the limitations of using a worst-case analysis.

**Relation To Broader Scientific Literature:**

As I was mentioning above, the paper technical content seem to be quite derivative compared to [1,2]. Nevertheless, the idea of investigating the tension between accuracy and robustness theoretically seem like a novel and interesting idea to investigate.

**Theoretical Claims:**

I checked the correctness of some of the proofs (proof of Theorems 3.1, 3.2, and 3.4) and only had a quick look over the rest of the proofs. Overall the proofs seem correct to me even-though they also seemed quite limited in terms of technical novelty.

In fact, the derivation of the lower and upper bounds on aggregation error appears to closely follow existing proofs in the literature see especially Section 8 in [1]. The Byzantine convergence analysis also resembles prior work (see e.g. [2] for lower bound and [1] for upper bound), and the paper does not sufficiently explain what modifications or adaptations were required. The connection of the convergence result with prior work is also not well explained in my opinion. Could the author explain what key insight we can get from Theorem besides that we have a lower bound in ($\frac{f}{n} G^2$), which is already what one has when considering the presence of Byzantine workers.

[1] Fixing by Mixing: A Recipe for Optimal Byzantine ML under Heterogeneity, Youssef Allouah, Sadegh Farhadkhani, Rachid Guerraoui, Nirupam Gupta, Rafael Pinot, John Stephan (AISTATS 2023)
[2] Byzantine-Robust Learning on Heterogeneous Datasets via Bucketing, Sai Praneeth Karimireddy, Lie He, Martin Jaggi (ICLR 2022)

---

> ### Author Rebuttal · Authors · 2025-03-30
>
> We sincerely thank the reviewer for their valuable time, insightful comments, and support of our work. We would like to answer the raised questions point by point as follows:
>
> **Q1.  I am unsure that the worst-case analysis is the best way to capture the accuracy vs robustness trade-off of Byzantine-robust methods. A more informative approach would be to analyze statistical bounds under assumed data distributions.**
>
> We agree with the reviewer that a statistical bound would be informative and provide a simple way to obtain a statistical bound based on the definition of $\epsilon$-accuracy below.
>
> When $x_1,\ldots,x_n$ are sampled from a probability distribution, we can take expectation on both sides of the inequality in Definition 2.2 ($\epsilon$-accuracy) and obtain that
> $$\mathbb{E}||Agg(x_1,\ldots,x_n)-\bar{x}||^2
> \leq \epsilon \cdot \mathbb{E}[\frac{1}{n}\sum_{i=1}^n ||x_i-\bar{x}||^2], $$
> where $\mathbb{E}[\frac{1}{n}\sum_{i=1}^n ||x_i-\bar{x}||^2]$ can be viewed as a measurement of the diversity. Specifically, when $x_1,\ldots,x_n$ are independently sampled from the same probability distribution with variance $\sigma^2$, we have
> $$\mathbb{E}||Agg(x_1,\ldots,x_n)-\bar{x}||^2
> \leq \epsilon\cdot \frac{n-1}{n}\sigma^2. $$
>
> Meanwhile, we would like to point out that there are some other measurements of diversity such as $\max_{i\neq j}\mathbb{E}||x_i-x_j||^2$ in existing works [2].  It still requires much more effort to study what statistics can be used to better analyze the Byzantine robustness, which we will leave for future work.
>
> **Q2. Clarify the novelty of the proof techniques and how they differ from prior work.**
> **Q2.(a). the derivation of the lower and upper bounds on aggregation error appears to closely follow existing proofs in the literature, especially Section 8 in [1].**
> We would like to point out politely that we only follow the notations (e.g., the notations of $f$ and $n$) in [1]. Although the format of our results may be similar to those in [1], the proof of bounds for aggregation error in this work is substantially different since we consider a different scenario without Byzantine workers.
>
> **Q2.(b).Could the author explain what key insight we can get from the Theorem besides that we have a lower bound in ($\frac{f}{n} G^2$), which is already what one has when considering the presence of Byzantine workers?**
> We would like to politely point out that when there are $f$ Byzantine workers, the lower bound should be $\frac{f}{n-2f}G^2$ (please refer to Table 1 in [1]), which can be infinitely large when $f\rightarrow(\frac{n}{2})_{-}$.
>
> Previous work [2] considers a special case where $\delta=\frac{f}{n}\leq\delta_{\max}<\frac{1}{2}$ and obtains a lower bound of the order $O(\delta)=O(\frac{f}{n})$. However, it cannot be extended to general cases of $\delta<\frac{1}{2}$ because $$\frac{f}{n-2f}=\frac{f}{n}\times\frac{n}{n-2f}=\frac{f}{n}\times\frac{1}{1-2\delta}\leq\frac{1}{1-2\delta_{\max}}\frac{f}{n}.$$ When $\delta_{\max}\rightarrow\frac{1}{2}$, the term $\frac{1}{1-2\delta_{\max}}$ will diverge to $+\infty$ and cannot be viewed as a constant any more.
>
> In summary, the lower bound $\frac{f}{n}G^2$ in this work will approach $\frac{1}{2}G^2$ when $f\rightarrow\frac{n}{2}$ (or equivalently, $\delta\rightarrow\frac{1}{2}$). On the contrary, the lower bounds in existing works considering the presence of Byzantine workers diverge to $+\infty$ when $f\rightarrow\frac{n}{2}$.
>
> **Q3. I would suggest including loss curves in the empirical evaluation.**
> We sincerely thank the reviewer for the constructive suggestion and will add the loss curves in the final version. Since we are not allowed to attach figures here, we present the test accuracy during the training process of using Multi-Krum when the hyper-parameter of Dirichlet distribution $\alpha=0.1$ in the following table. The added empirical results further support the conclusion of our work.
>
> |Epoch|19|39|59|79|99|119|139|159(final)|
> |-|-|-|-|-|-|-|-|-|
> |$f=0$|77.27%|84.70%|84.46%|86.90%|87.81%|88.82%|89.17%|89.42%|
> |$f=1$|55.04%|78.78%|83.64%|84.55%|86.36%|87.27%|87.77%|88.05%|
> |$f=3$|45.24%|72.90%|78.00%|81.11%|82.60%|82.19%|82.98%|83.50%|
> |$f=5$|30.41%|37.60%|52.83%|63.48%|69.01%|69.55%|69.68%|69.86%|
> |$f=7$|18.68%|25.55%|27.64%|30.88%|33.98%|38.72%|40.05%|40.31%|
>
> **Q4. They do not seem to explain very clearly the behavior of state-of-the-art methods like NNM or Bucketing.**
> To answer the question, we provide another perspective of viewing NNM and bucketing below. An aggregator combined with NNM (or bucketing) can be considered as a new aggregator that can resist fewer Byzantine workers but have less aggregation error. Therefore, when using NNM or bucketing, we actually use the prior knowledge of the Byzantine worker number to make a better trade-off between robustness and accuracy.
>
> ----
> We hope that our response can address the reviewer's concerns, and we greatly thank the reviewer again for the support of our work.

---

### Official Review · Reviewer_RMMi · 2025-03-13

**Overall Recommendation:** 4

**Summary:**

The paper analysis the learning error in distributed learning induced by robust aggregation schemes in the case when the actual number of Byzantine workers is 0, while the system is designed to handle a non-zero number of Byzantine workers $f$. The paper makes important contributions to the field of robustness in distributed learning by re-analyzing the errors of SOTA robust aggregation in the above-mentioned setting (i.e., no actual Bzyantine workers but non-zero $f$). The obtained bounds are better than those that assume presence of Byzantine workers, especially when $f$ approaches the limit $\frac{n}{2}$. The presented analysis is useful for studying the impact of Byzantine-robustness in ideal scenarios when the actual number of Byzantine workers (in most learning rounds) are much smaller than the maximum possible number of Byzantine workers (in any given learning round). The paper can have long lasting implications.

**Claims And Evidence:**

Yes.

**Essential References Not Discussed:**

Not that I could think of.

**Experimental Designs Or Analyses:**

While I did not reproduce the experimental results, they appear sound/valid.

**Methods And Evaluation Criteria:**

Yes.

**Other Comments Or Suggestions:**

Some suggestions for future extensions:
1. Utility of the obtained results for studying the algorithmic stability (thereby, generalization power) of ByzGD.
2. Interpolation between the two extreme cases: i) actual number of Byzantine worker is 0 and ii) actual number of Byzantine workers is $f$.
3. Did you mean to present Theorem 4.6 for $\epsilon-$accurate aggregation rule? Lines 328 - 329 after the theorem mentions $\epsilon G^2$ and $\frac{\epsilon G^2}{2 \mu}$ terms.

A few typos:
1. Line 110: For 'an' $(f, \kappa)-$robust ...

**Other Strengths And Weaknesses:**

Strengths:
1. Good comprehensible proofs with proper explanation and motivation between steps.
2. The proof techniques can be applied to future research in this field.

Weaknesses:
1. Impact on learning error under the more general heterogeneous setting of (G, B)-dissimilarity is missing.

**Questions For Authors:**

See my suggestions and weaknesses above.

**Relation To Broader Scientific Literature:**

The paper makes important contributions to an important field of Byzantine-robustness in distributed learning.

**Theoretical Claims:**

Yes. I (quickly) checked the proofs of Theorems 3.1, 3.2 and 3.6. I did not find any issues.

---

> ### Author Rebuttal · Authors · 2025-03-28
>
> We sincerely thank the reviewer for their valuable time, constructive suggestions, and support of our work. We would like to respond point by point below.
>
> **Comment 1. Impact on learning error under the more general heterogeneous setting of (G, B)-dissimilarity is missing.**
>
> We agree with the reviewer that analyzing the tension under the more general setting of $(G, B)$-dissimilarity can help improve the theoretical contribution. Meanwhile, we politely think that adding the theory under $(G, B)$-dissimilarity requires substantial improvement, and the current version already makes an important contribution, as the reviewer mentioned. We will study the learning error under more general settings in future work and sincerely thank the reviewer again for the constructive suggestion.
>
> **Comment 2. Suggestions: (a) Utility of the obtained results for studying the algorithmic stability (thereby, generalization power) of ByzGD; (b) Interpolation between the two extreme cases: i) actual number of Byzantine worker is 0 and ii) actual number of Byzantine workers is $f$.**
>
> We greatly thank the reviewer for the constructive suggestions and will study the two problems in future work.
>
> **Comment 3. Did you mean to present Theorem 4.6 for $\epsilon$-accurate aggregation rule?**
>
> We sincerely thank the reviewer for pointing out the typo. Theorem 4.6 for $\epsilon$-accurate (instead of $(f,\kappa)$-robust) aggregation rule. Please refer to Appendix A.6 for the correct result. We will fix this typo in the maintext in the final version.
>
> Meanwhile, we would clarify that the novelty and the contribution of this paper will be almost not affected by the revision. Specifically, as mentioned in our response to Concern 1, the main contributions of this paper are the lower bounds. Please note that $\epsilon\geq\frac{f}{n-f}$ (Theorem 3.1) and that $TM_{f/n}$ is both $(f,\kappa)$-robust and $\epsilon$-accurate with $\epsilon=\frac{f}{n-f}$. It shows the tightness of the lower bound in Theorem 4.5.
>
> **Comment 4. A few typos ...**
> We greatly thank the reviewer for pointing out the typos. We promise to proofread the script carefully and fix the typos in the final version.
>
> We sincerely thank the reviewer again for the great support of our work, and we are always willing to answer any further questions.

---

> > ### Comment · Reviewer_RMMi · 2025-04-02
> >
> > I thank the authors for responding to my comments. I will keep my score.

---

> > > ### Author Response · Authors · 2025-04-02
> > >
> > > We sincerely thank the reviewer for the quick acknowledgement and the support of our work.

---

### Official Review · Reviewer_rsEj · 2025-03-17

**Overall Recommendation:** 3

**Summary:**

This paper examines distributed learning in a setting where the server implements a robust aggregation rule. Motivated by the Byzantine-robust learning framework, it evaluates the performance of distributed gradient descent (GD) methods designed to cope with Byzantine workers, even when none are present. The authors extend the definition of $(\delta,\kappa)$-robustness to scenarios without Byzantine workers by introducing the notion of $\epsilon$-accuracy, derive accuracy coefficients for several robust aggregation rules, establish a convergence result and a lower bound, and support their findings with numerical experiments on CIFAR-10 that illustrate the impact of varying the "assumed" number of Byzantine workers.

**Claims And Evidence:**

Yes

**Essential References Not Discussed:**

N/A

**Experimental Designs Or Analyses:**

N/A

**Methods And Evaluation Criteria:**

Yes

**Other Comments Or Suggestions:**

In the proof of Theorem 4.6, the term in the upper bound related to heterogeneity (for both non-convex and PL functions) is proportional to $\frac{f}{n-f}$, whereas according to the proof (in Appendix A.6) it should be instead proportional to $\epsilon$ (or $\kappa$) -- the accuracy of the actual aggregator used and not the optimal accuracy.

**Other Strengths And Weaknesses:**

- The experimental setup is sound and detailed.
- The analysis is done for ByzGD, where local gradients are computed exactly. While the authors mention ByzSGD as a variant, it is in fact much more common and brings up additional challenges: even in homogeneous scenarios (when $G=0$), simply applying a robust aggregation rule isn’t sufficient—even when Byzantine workers are absent—as seen in cases with skewed noise distributions (e.g., Counterexample 3 and others in [2]). They suggest that incorporating momentum or a similar strategy that leverages historical gradient data is necessary for true Byzantine-robustness, yet no analysis for momentum is provided (considering just SGD without momentum won't work in my opinion).

**Questions For Authors:**

N/A

**Relation To Broader Scientific Literature:**

The proof of Theorem 4.6 closely follows the proof of Theorem 1 in [1], with Definition 2.2 replacing Definition 2.1 (which is expected, given that the definitions coincide when $f=0$, as the authors mention). Therefore, the analysis is of limited novelty as it can be directly deduced from [1].


[1] Allouah, Farhadkhani, Guerraoui, Gupta, Pinot, Stephan. "Fixing by Mixing: A Recipe for Optimal Byzantine ML under Heterogeneity", AISTATS, 2023.

**Theoretical Claims:**

I checked the correctness of the proofs for the upper and lower bounds (Theorems 4.5 and 4.6) .

---

> ### Author Rebuttal · Authors · 2025-03-28
>
> We sincerely thank the reviewer for the valuable time and the detailed review. We will respond to the raised concerns point by point as follows:
>
> **Concern 1. The proof of Theorem 4.6 is of limited novelty as it can be directly deduced from [1].**
> We thank the reviewer for letting us know their concern and would like to clarify the meaning of Theorem 4.6 below.
>
> + We would like to first politely point out that the notation $f$ in [1] is the **known** Byzantine worker number, while $f$ in this work is the **assumed** Byzantine worker number. To the best of our knowledge, this is the first convergence result for ByzGD with any $(f,\kappa)$-robust aggregator when there are actually no Byzantine workers.
> + Meanwhile, since this work mainly focuses on the tension between Byzantine robustness and no-attack accuracy, the main contribution of Section 4 lies in the lower bound (i.e., Theorem 4.5). The main purpose of presenting Theorem 4.6 here is to show the tightness of Theorem 4.5.
>
> **Concern 2. The analysis is done for ByzGD and mentions ByzSGD as a variant. However, considering just SGD without momentum won't work in my opinion.**
> We thank the reviewer for the insightful comment. As discussed in Section 4 (please refer to the right column of line 326), using momentum can reduce the variance (which the aggregation error is proportional to) in ByzSGD. Thus, ByzSGD with momentum has a better convergence rate and more Byzantine robustness than ByzSGD. However, compared to ByzGD, ByzSGD with momentum does not improve the theoretical learning rate w.r.t. the iteration number $T$.
>
> For example, we consider ByzGD with momentum as a special case of ByzSGD with momentum (with variance $\sigma^2=0$). For the same case as in line 314 to line 320 where $$F_1(w)=\ldots=F_f(w)=\frac{nG}{2\sqrt{f(n-f)}}w^2$$ and $$F_{f+1}(w)=\ldots=F_n(w)=\frac{nG}{2\sqrt{f(n-f)}}(w^2-2w),$$ the aggregated result will be totally controlled by worker $\{f+1,\ldots,n\}$, and from the server's perspective, it is equivalent to optimize the objective $\tilde{F}(w)=\frac{nG}{2\sqrt{f(n-f)}}(w^2-2w)$, whatever the optimizer (e.g. ByzGD with momentum) is used. Thus, ByzSGD with momentum cannot theoretically improve over ByzGD in this case.
>
> Additionally, we failed to find reference [2] in the review although the reviewer mentioned it. We politely guess that the reference might be [B], which is presented below. Please correct us if the guess is wrong.
>
> [B] Karimireddy, S. P., He, L., and Jaggi, M. Learning from history for Byzantine robust optimization. In Proceedings of the International Conference on Machine Learning, pp. 5311–5319, 2021.
>
>
> **Concern 3. In the proof of Theorem 4.6, the term in the upper bound related to heterogeneity (for both non-convex and PL functions) is proportional to $\frac{f}{n-f}$, whereas according to the proof (in Appendix A.6) it should be instead proportional to $\epsilon$ (or $\kappa$) -- the accuracy of the actual aggregator used and not the optimal accuracy.**
>
> We sincerely thank the reviewer for pointing out the typo, and we will fix this typo in the final version.
>
> Meanwhile, we would clarify that the novelty and the contribution of this paper are almost not affected by the revision. Specifically, as mentioned in our response to Concern 1, the main contributions of this paper are the lower bounds. Please note that $\epsilon\geq\frac{f}{n-f}$ (Theorem 3.1) and that $TM_{f/n}$ is both $(f,\kappa)$-robust and $\epsilon$-accurate with $\epsilon=\frac{f}{n-f}$. It shows the tightness of the lower bound in Theorem 4.5.
>
> **Additional Clarification**
> Finally, we would like to restate the main theoretical contributions of this work briefly. In this paper, we provide lower bounds for the aggregation error (Section 3) and convergence rate of ByzGD (Section 4) for $(f,\kappa)$-robust aggregators. Moreover, we show the tightness of the lower bounds.
>
> We sincerely thank the reviewer again for the insightful review and hope that our response can address the reviewer's concerns. Meanwhile, we would greatly appreciate it if the reviewer could re-evaluate our work in light of our response.

---

> > ### Comment · Reviewer_rsEj · 2025-04-03
> >
> > I thank the authors for their clarification and for addressing my concerns. I understand that the main contribution is the lower bound in Theorem 4.5, and I agree with the authors that this result also applies to the stochastic case. While deriving an upper bound for ByzSGD with momentum (providing the **rate** of convergence to an $\epsilon G^2$-order neighborhood) would be interesting given its practical relevance, I no longer view its absence as a significant drawback of this work. I have updated my rating accordingly.
> >
> > I have one small question regarding the distinction between $(f,\kappa)$-robustness and $\epsilon$-accuracy. From Tables 1 and 2, it appears that although TM has a strictly suboptimal $\kappa$, it achieves optimal $\epsilon$-accuracy. Can the authors please elaborate on this discrepancy? I notice that the analysis differs from that of Allouah et al. (2023).

---

> > > ### Author Response · Authors · 2025-04-04
> > >
> > > We sincerely thank the reviewer for the support of our work and the insightful follow-up comment. Our response to the follow-up question regarding the distinction between $(f,\kappa)$-robustness and $\epsilon$-accuracy is presented below.
> > >
> > > Firstly, the $(f,\kappa)$-robustness property measures the worst-case aggregation error in the presence of Byzantine workers, while $\epsilon$-accuracy measures the worst-case aggregation error without Byzantine workers. We politely think that for some specific aggregator such as TM, an optimal $\epsilon$ does not necessarily lead to an optimal $\kappa$.
> > >
> > > Secondly, in Section 8.2 of the Appendix of Allouah et al. (2023), it says that the $\kappa$ values are tight in order of magnitude. However, it is uncertain whether the numerical constants are tight. The numerical constants may be further optimized. Meanwhile, we think that further optimization of $\kappa$ values is a challenging but interesting direction for future work.
> > >
> > > We thank the reviewer again for their valuable time, support of our work, and insightful comments. We promise to take all the reviews into consideration and revise accordingly in the final version.

---

### Official Review · Reviewer_k8Cc · 2025-03-19

**Overall Recommendation:** 4

**Summary:**

This work studies robust aggregation methods in the Byzantine setting. Specifically, let $x_i \in \mathbb{R}^d$ be information held by worker $i$, and suppose that the goal is to compute the mean $\frac{1}{n}\sum_{i=1}^n x_i$. In the Byzantine setting, an unknown subset of $f$ workers are adversarially corrupted, and thus a robust aggregator $Agg(x_1, \dots, x_n)$ is used to approximate the mean of the uncorrupted workers' vectors, i.e., $\frac{1}{n-f}\sum_{i \in S} x_i$, where $S$ denotes the unknown set of uncorrupted workers. Such aggregators aim to approximate this quantity reasonably accurately without knowing $S$ in advance.

While prior work has developed methods for robust aggregation, this work considers the scenario where previously proposed methods are applied even though there are actually no Byzantine workers present. This serves as a sanity check measuring the "price of robustness." To quantify robustness, they define an aggregator to be $(f,\kappa)$-robust if it approximates the desired output within a factor of $\kappa$ times the variance of the inputs, for every subset of uncorrupted workers of size at least $n-f$. To measure accuracy, they say an aggregator is $\epsilon$-accurate if its estimate is within $\epsilon$ times the variance of the inputs from the true mean of all $n$ workers.

In their first set of results, they derive upper and lower bounds relating the accuracy $\epsilon$ of an estimator to its robustness parameter $\kappa$. They show general bounds satisfying $\frac{f}{n - f} \leq \epsilon \leq \kappa$, along with tighter, method-specific bounds for particular estimators (namely the geometric median, coordinate-wise trimmed mean, and coordinate-wise median). In all cases, they demonstrate that their bounds are relatively tight by explicitly constructing examples where these bounds are attained.

Then, they turn their attention to one of the most common use cases of Byzantine aggregation, namely the aggregation of gradients. In this setting, each worker has its own local dataset and computes a gradient $\nabla F_i(w)$, where $w$ denotes the current shared model parameters and $F_i$ represents the local loss function evaluated on worker $i$'s dataset.

In this setting, under several (relatively standard) assumptions on the loss functions, they provide lower and upper bounds for how closely $T$ steps of Byzantine gradient descent (where each step updates $w$ based on a robust aggregation of gradients computed by the workers) approximate the optimal loss. Specifically, their lower bounds hold under the assumption that the Polyak–Łojasiewicz condition is satisfied, indicating that achieving near-optimal loss through standard gradient descent would, in principle, be feasible. In their upper bound, they explicitly show that the accuracy term $\frac{f}{n - f}$, previously derived for general Byzantine aggregators, directly affects how closely one can approach the optimal loss.

Empirically, the authors validate their theoretical results by training a ResNet-20 model on CIFAR-10 using Byzantine gradient descent with robust aggregators, specifically multi-Krum and coordinate-wise trimmed mean. They vary the maximum number of Byzantine workers ($f$) the aggregators can tolerate and the degree of heterogeneity in data distributions across workers (controlled by the Dirichlet parameter $\alpha$). Their results demonstrate that even in the absence of actual Byzantine attacks, increasing the robustness parameter $f$ significantly reduces accuracy on the clean test set, with this effect being particularly pronounced when data distributions across workers are highly heterogeneous (small $\alpha$). These empirical findings confirm their theoretical claim of an inherent tradeoff between robustness to Byzantine workers and accuracy under no attack.

**Claims And Evidence:**

Yes. They prove their claims and their experiments make sense.

**Essential References Not Discussed:**

Not to my knowledge.

**Experimental Designs Or Analyses:**

The experiment design made complete sense to me. It didn't require significant checking due to its very simple design (i.e. simply measuring the effectiveness of byazantine algorithms with varying levels of tolerance and heterogeneity).

**Methods And Evaluation Criteria:**

Yes, especially given that their contributions are primarily theoretical. I view the experiments a basic sanity check to illustrate the effect of their theorems in practice.

**Other Comments Or Suggestions:**

None.

**Other Strengths And Weaknesses:**

Strengths: i think the theory is simple, easy to follow, and also quite illustrative. I generally like the theme of understanding tension between accuracy and robustness (which appears in many different fields of machine learning!)

Weaknesses: Some of the results on specific aggregation methods seem like they could be differed to the appendix (given that they are quite simple). I find the general bounds more interesting and would be interested in further theoretical analysis of these bounds in contexts of explicit heterogeneity (i.e. assume $x_i$ are sampled from some distirbution, etc. etc.).

**Questions For Authors:**

What are some future directions for investigating this problem under more concrete assumptions on the hetrogeneity of the data? It feels like in those situations, one might hope for a "best of both worlds" by coming up with some sort of scheme to detect if one is in the byzantine setting or not. It also feels like there might be some more interesting theoretical bounds one could come up with under more structural assumptions on the data.

**Relation To Broader Scientific Literature:**

They seem well related. This does appear to be the first work that sanity checks existing methods on uncorrupted data, and this paper does well in positioning itself as such.

**Theoretical Claims:**

I scanned through the proofs and they appear correct. They also all make very intuitive sense and so i do not doubt their validity.

---

> ### Author Rebuttal · Authors · 2025-03-26
>
> We sincerely thank the reviewer for the insightful comments, the constructive suggestions, and the support of our work. We would like to respond to the raised questions point by point below.
>
> **Comment 1. Some of the results on specific aggregation methods could be deferred to the appendix.**
> In the current version, we present the definitions and results of several common robust aggregation methods (GM, TM, and CM) in order to be more friendly to the readers who are not very familiar with Byzantine-robust distributed learning. Meanwhile, we agree with the reviewer that part of these texts could be deferred to the appendix for better readability. We will revise it in the final version and thank the reviewer for the constructive suggestion.
>
> **Comment 2. I find the general bounds more interesting and would be interested in further theoretical analysis of these bounds in contexts of explicit heterogeneity (i.e. assume $x_i$ are sampled from some distribution, etc. etc.).**
> For the case where $x_i$ are independently sampled from the same distribution with a variance of $\sigma^2$, we can take expectation on the inequality in Definition 2.2 (line 126). According to (15) on page 13 in the Appendix and using the fact that $x_i$ and $x_j$ are independent $(i\neq j)$, we can obtain that
> $$\mathbb{E}||Agg(x_1,\ldots,x_n)-\bar{x}||^2\leq \epsilon \cdot \mathbb{E}[\frac{1}{n}\sum_{i=1}^n ||x_i-\bar{x}||^2]=\epsilon\cdot \frac{n-1}{n}\sigma^2. $$
> Meanwhile, we will also investigate more cases in future work and sincerely thank the reviewer for the suggestion.
>
> **Comment 3. What are some future directions for investigating this problem under more concrete assumptions on the hetrogeneity of the data?**
> Some potential future directions are presented below:
> + As pointed out by reviewer RMMi, investigating the tension between Byzantine robustness and no-attack accuracy under the more general $(G, B)$-dissimilarity assumption is a direction of future extension.
> + Additionally, reviewer RMMi also provides a potential future direction of studying the tension when the actual number of Byzantine workers is larger than $0$ but smaller than $f$.
> + The analysis of this work is for general cases, and the bounds are related to the case with an extreme large skeness. In some real-world applications, the distribution of training instances and gradients is usually not that extreme. Thus, we find it will be a potential future direction to investigate the data distribution in real-world applications and analyze the tension for some specific distributions.
>
> **Comment 4. It feels like in those situations, one might hope for a "best of both worlds" by coming up with some sort of scheme to detect if one is in the byzantine setting or not. It also feels like there might be some more interesting theoretical bounds one could come up with under more structural assumptions on the data.**
> We agree with the reviewer's insightful comments, which inspire us to think more about future directions. In this paper, we mainly investigate the tension and prove the tightness of our results for general cases. In real-world applications, as the reviewer pointed out, we could utilize the observation of some true data. Under the assumption that the non-Byzantine data is close to the observed data, we may obtain a better result about the tension between Byzantine robustness and no-attack accuracy. Since we mainly focus on the general cases, it is beyond the scope of this paper. We will explore this in future work and sincerely thank the reviewer for the constructive comments.

---

> > ### Comment · Reviewer_k8Cc · 2025-04-05
> >
> > Thank you for your rebuttal. I will maintain my (positive) score and view of this paper.

---

> > > ### Author Response · Authors · 2025-04-05
> > >
> > > Thank you once again for your acknowledgement, constructive suggestions and support of our work. We will take all the reviews into consideration and make revisions accordingly in the final version.

---

### Decision · Program_Chairs · 2025-05-01

**Decision:**

Accept (spotlight poster)

**Comment:**

This paper investigates the trade-off between Byzantine robustness and attack-free learning error in distributed learning. To be specific, it analyzes the aggregation errors of robust aggregation rules, as well as the learning error of Byzantine-robust gradient decent equipped with these robust aggregation rules, both in the absence of Byzantine workers. The theoretical results show that making Byzantine-robust gradient decent robust to more Byzantine workers will increase the worst-case attack-free learning error. Although the theoretical analysis resembles those from the existing works and limited to gradient descent (not stochastic gradient descent), this paper provides novel insights into the tension between Byzantine robustness and attack-free learning error. All reviewers unanimously recommend acceptance.